# Low frequency variants associated with leukocyte telomere length in the Singapore Chinese population

Xuling Chang [1,2], Resham L. Gurung[3], Ling Wang[4], Aizhen Jin[5], Zheng Li [4], Renwei Wang[6], Kenneth B. Beckman [7], Jennifer Adams-Haduch[6], Wee Yang Meah[4], Kar Seng Sim [4], Weng Khong Lim [8,9,10], Sonia Davila[8,11], Patrick Tan [4,8,9,12], Jing Xian Teo[8], Khung Keong Yeo [8,13], Yiamunaa M.[3], Sylvia Liu[3], Su Chi Lim [3,14,15], Jianjun Liu [4,16], Rob M. van Dam[15,16], Yechiel Friedlander[17], Woon-Puay Koh[5,15], Jian-Min Yuan [6,18], Chiea Chuen Khor [4,19], Chew-Kiat Heng [1,2,20] & Rajkumar Dorajoo [4,5,20]

The role of low frequency variants associated with telomere length homeostasis in chronic diseases and mortalities is relatively understudied in the East-Asian population. Here we evaluated low frequency variants, including 1,915,154 Asian specific variants, for leukocyte telomere length (LTL) associations among 25,533 Singapore Chinese samples. Three East Asian specific variants in/near POT1, TERF1 and STN1 genes are associated with LTL (Meta-analysis $P$ $2.49 \times 10^{-14}$–$6.94 \times 10^{-10}$). Rs79314063, a missense variant (p.Asp410His) at POT1, shows effect 5.3 fold higher and independent of a previous common index SNP. TERF1 (rs79617270) and STN1 (rs139620151) are linked to LTL-associated common index SNPs at these loci. Rs79617270 is associated with cancer mortality [HR$_{95\%CI}$ = 1.544 (1.173, 2.032), $P_{Adj}$ = 0.018] and 4.76% of the association between the rs79617270 and colon cancer is mediated through LTL. Overall, genetically determined LTL is particularly associated with lung adenocarcinoma [HR$_{95\%CI}$ = 1.123 (1.051, 1.201), $P_{adj}$ = 0.007]. Ethnicity-specific low frequency variants may affect LTL homeostasis and associate with certain cancers.

[1] Department of Paediatrics, Yong Loo Lin School of Medicine, National University of Singapore, Singapore, Singapore. [2] Khoo Teck Puat—National University Children's Medical Institute, National University Health System, Singapore, Singapore. [3] Clinical Research Unit, Khoo Teck Puat Hospital, Singapore, Singapore. [4] Genome Institute of Singapore, Agency for Science, Technology, and Research, Singapore, Singapore. [5] Health Services and Systems Research, Duke-NUS Medical School Singapore, Singapore, Singapore. [6] Division of Cancer Control and Population Sciences, UPMC Hillman Cancer Center, University of Pittsburgh, Pittsburgh, PA, USA. [7] University of Minnesota Genomics Center, University of Minnesota, Minneapolis, MN, USA. [8] SingHealth Duke-NUS Institute of Precision Medicine, Singapore, Singapore. [9] Cancer & Stem Cell Biology Program, Duke-NUS Medical School, Singapore, Singapore. [10] SingHealth Duke-NUS Genomic Medicine Centre, Singapore, Singapore. [11] Cardiovascular and Metabolic Disorders Program, Duke-NUS Medical School, Singapore, Singapore. [12] Cancer Science Institute of Singapore, National University of Singapore, Singapore, Singapore. [13] Department of Cardiology, National Heart Centre Singapore, Singapore, Singapore. [14] Diabetes Centre, Admiralty Medical Centre, Singapore, Singapore. [15] Saw Swee Hock School of Public Health, National University of Singapore, Singapore, Singapore. [16] Department of Medicine, Yong Loo Lin School of Medicine, National University of Singapore, Singapore, Singapore. [17] School of Public Health and Community Medicine, Hebrew University of Jerusalem, Jerusalem, Israel. [18] Department of Epidemiology, Graduate School of Public Health, University of Pittsburgh, Pittsburgh, PA, USA. [19] Singapore Eye Research Institute, Singapore National Eye Centre, Singapore, Singapore. [20]These authors jointly supervised this work: Chew-Kiat Heng, Rajkumar Dorajoo. ✉email: paehck@nus.edu.sg; dorajoor@gis.a-star.edu.sg

Telomere length has been associated with risks of cancers, other chronic diseases and mortalities[1–4]. Shortening of telomere length leads to an increased risk of age-related diseases such as cardiovascular disease[1,2,5]. When telomere length is critically shortened, cells enter a state of growth arrest[6,7]. Such growth-arrested cells may remain in a state of cellular senescence for long periods of time, a phenomenon now also recognized as a likely tumor suppressor mechanism[8]. In contrast, the enzyme telomerase, which lengthens telomeres and promotes cell proliferation and survival, is activated in most human cancers[3,9]. Epidemiological examinations on the relationship between telomere length and carcinogenesis have been inconsistent, with both longer and shorter telomere length associated with cancer risks, even among the same type of cancer[10]. These discrepancies may be in part due to reverse causation[10]. Genetic studies, utilizing germline variants as proxies for telomere length, have recently indicated the potential causal effect between longer leukocyte telomere length (LTL) and increased risks of several cancers[1].

Unraveling genetic factors regulating LTL could provide insight into telomere homeostasis. LTL are highly heritable (~30–60%)[11,12]. Our previous genome-wide association study (GWAS) in the Singapore Chinese population had identified 10 common variants, with minor allele frequency (MAF) > 1%, associated with LTL[4]. We also found East Asian-specific genetic predispositions to increased telomere attrition, for example, at the *TERF1* locus[4]. Genetic studies performed in the European, South Asian and African populations have identified 12 other common loci associated with LTL and further highlighted on ethnic disparities[13–15]. Besides common variants, low-frequency variants (MAF < 1%) may also play an important role in accelerated LTL attrition. A recent study suggested that low-frequency variants of genes involved in telomere maintenance and ageing, such as *ATM* and *RPL8* genes, may affect LTL in Europeans, although additional replications are necessary for robust confirmations[16]. However, little is known if such rare variants involved in telomere homeostasis exist in the East Asian population and if these in turn affect predispositions to chronic diseases and mortalities.

The 1000 Genomes cosmopolitan reference panels for imputation[17] have significantly improved genomic coverage in GWAS analyses, but have been largely powered to evaluate common variants (MAF > 1%). Alternatively, studies have shown that utilizing population-specific reference panels or in combination with 1000 Genome reference panels during imputation, significantly improved accuracy of low-frequency variants (MAF < 1%) for the relevant study population[18–21].

We present a genetic study for LTL associations in the East Asian Singapore Chinese population focused on low-frequency variants (MAF 0.05%–1%), ascertained through imputation procedures that combined local Singapore population-specific reference panels[22] with the 1000 Genomes reference panels. Subsequently, LTL-associated variants were evaluated for associations with mortality and incident cancers.

## Results

### Genome-wide LTL-associated variants in the Singapore Chinese.
Our common variant (MAF > 1%) analysis utilizing the two panel imputation procedure in the Singapore Chinese Health Study (SCHS) Discovery dataset recapitulated all ten genome-wide hits we had previously reported[4] (Supplementary Fig. 2), with the same index SNPs or those in linkage disequilibrium [(LD), $r^2 > 0.6$ 1000 Genomes EAS panel] to reported index SNPs (Supplementary Figs. 3 and 5). At the *POT1* gene region, we identified an additional common SNP, rs34398311 ($P = 3.23 \times 10^{-13}$, Supplementary Fig. 5) but this did not show strong independent statistical association (Supplementary Table 5).

Inclusion of low-frequency variants (MAF between 0.05% and 1%) improved LTL heritability estimates in our study and SNP-based $h^2$ of low-frequency variants was ~2.9% in the SCHS discovery dataset (Supplementary table 6). Four genome-wide significant low-frequency variants for LTL ($P$ between $9.49 \times 10^{-13}$–$1.17 \times 10^{-8}$, Table 1, Fig. 1) were identified in the SCHS Discovery dataset. All four variants were monomorphic in non-East Asian reference populations, while MAFs in East Asian reference populations from 1000 Genomes and GnomAD[23,24] were similar to that in the SCHS (0.6%–0.8%).

Two low-frequency signals were observed at the *POT1* gene locus on chromosome 7. *POT1* rs79314063 (p.Asp410His) and rs186524197 (intergenic) were in LD (SCHS $r^2 = 0.883$, Supplementary Fig. 6) and conditional probability analyses in the SCHS dataset indicated that the associations identified at these two variants were linked (Supplementary Table 7). Rs79314063 was a genotyped missense variant that displayed strong deleterious functional effects (Supplementary Table 8) and strong CADD annotation score (scaled CADD = 22.5, top 1% of deleterious variants in the human genome [The minor (G) allele was associated with significantly shorter LTL ($P = 6.94 \times 10^{-10}$)]). The change from Asp to His residue at p.410 in POT1 indicated a relatively high deleterious effect score of 73 (85% accuracy) using SNAP2[25]. The putative mutant protein containing the His amino acid at p.410, did not affect solvent accessibility of residues at the OB-fold and stacking residues of POT1. However, the His amino acid at p.410 modified solvent accessibility of 17 out of 305 residues within the TPP1 interacting domain as compared to the wildtype protein (p.391, 396, 413, 417, 419, 451, 196, 534 and 575 were changed to exposed while p.336, 399, 418, 440, 553, 592, 504 and 628 were changed to buried with the His residue at p.410) (Supplementary Table 9). Rs79314063 was also independent from the reported common index SNP at the *POT1* locus identified in our previous study (rs7776744)[4] ($r^2 < 0.05$ 1000 Genomes EAS panel and Supplementary Table 10). The low-frequency variant (rs79314063) also displayed much stronger LTL effects that were 5.3-fold higher than the effect of the reported common index variant at *POT1*[4] (Supplementary Table 11).

The two other low-frequency variants, rs79617270 (chromosome 8) and rs139620151 (chromosome 10) were also close to identified common index SNPs from our previous East Asian GWAS study [common index SNPs, rs28365964 at the *TERF1* locus in chromosome 8 and rs12415148 at the *STN1* locus (also known as *OBFC1*) in chromosome 10][4]. The identified low-frequency variants at *TERF1* and *STN1* loci exhibited stronger effects as compared to the common index SNPs at these loci (~1.4 and ~1.7-fold higher, respectively) (Supplementary table 11). Adjusting the low-frequency variants effects on the allele dosage of previously identified common SNPs at the loci, resulted in attenuation of statistical significance ($P$ between 0.007–0.017, Supplementary Table 10), suggesting that low frequency and common SNP signals at these two regions were not independent of one another.

Rs79617270 was located within the 3' untranslated region of the *SBSPON* gene. Chromatin interaction mapping indicated that a functional genetic element containing this variant significantly interacts with *TERF1* and *SBSPON* in various cell types, including blood lymphoblast (GM12878) (FDR < $5.94 \times 10^{-9}$, Supplementary Table 12 and Supplementary Fig. 7). Rs139620151 is an intronic variant in *SH3PXD2A* and close to the *STN1* gene. Chromatin interaction mapping indicated that this intronic region at *SH3PXD2A* may affect proximal *STN1* and *GSTO2* genes in lung fibroblast cells (IMR90) (FDR < $3.99 \times 10^{-7}$, Supplementary Table 12 and Supplementary Fig. 8).

### Validation of low-frequency variants associated with LTL. 
We replicated low-frequency variants identified in the discovery study

**Table 1 Summary statistics of low-frequency genome-wide hits identified in the study.**

| SNP | Chr | Position | Genes | TA | TAF | Discovery N = 20,177 | | | Replication N = 5356 | | | Meta-analysis N = 25,533 | | |
|---|---|---|---|---|---|---|---|---|---|---|---|---|---|---|
| | | | | | | Beta | Se | P | Beta | Se | P | Beta | Se | P |
| rs18524197 | 7 | 124431653 | POT1 | A | 0.007 | −0.360 | 0.059 | $1.32 \times 10^{-9}$ | In linkage disequilibrium with rs79314063 ($r^2 = 0.883$). | | | | | |
| rs79314063 | 7 | 124481168 | POT1 | G | 0.007 | −0.345 | 0.057 | $1.15 \times 10^{-9}$ | −0.191 | 0.131 | 0.146 | −0.320 | 0.052 | $6.94 \times 10^{-10}$ |
| rs79617270 | 8 | 73978144 | TERF1/SBSPON | G | 0.008 | 0.384 | 0.054 | $9.49 \times 10^{-13}$ | 0.328 | 0.123 | 0.008 | 0.376 | 0.049 | $2.49 \times 10^{-14}$ |
| rs139620151 | 10 | 105593428 | STN1/SH3PXD2A | A | 0.006 | 0.383 | 0.067 | $1.17 \times 10^{-8}$ | 0.456 | 0.163 | 0.005 | 0.394 | 0.062 | $2.16 \times 10^{-10}$ |

*Chr chromosome, TA test allele, TAF test allele frequency.*

in four independent datasets (N = 5356). All three variants identified from the discovery GWAS showed directionally consistent associations in the individual replication datasets and for rs79617270 and rs139620151, statistically significant associations were observed (P between 0.005–0.008, Table 1 and Supplementary Table 13). Although the missense variant (rs79314063) in the POT1 gene did not show statistically significant association with LTL in the replication datasets alone (P = 0.146), the replication data improved overall meta-analysis association levels (Table 1, Meta P = $6.94 \times 10^{-10}$). Additionally, we evaluated identified variants in a separate Singaporean Chinese dataset with WGS data (N = 154). The variants showed similarly consistent direction of effects in this dataset and rs139620151 was statistically significant (P = 0.009, Supplementary Table 14). Further adjustment for smoking status did not affect the LTL associations of these SNPs (Supplementary table 15).

**Replication of known LTL-associated low-frequency variants**. Rare variants in ATM and RPL8 were associated with LTL in Europeans[16]. Rs529546508 in RPL8 was monomorphic in East Asians and the only low-frequency variant within this gene captured in the SCHS, rs763648407, was not associated with LTL (P = 0.332, Supplementary Table 16). Variants reported at ATM were rare in Europeans but more common in the SCHS (MAF > 1%) and were not associated with LTL (Supplementary Table 16). However, we observed an East Asian-specific rare missense variant, rs3218670 (MAF = 0.085%), at ATM that was nominally associated with LTL in our dataset (P = 0.003, Supplementary Table 16). The POT1 p.V326A variant, recently identified in the Japanese population was not tested in the present study as it did not pass imputation info score threshold (info score between 0.094 and 0.625)[26]. Gene burden tests using all potentially deleterious coding variants also indicated only POT1 as robustly significant and these effects were driven by rs79314063 (Supplementary Table 17).

**Associations with mortality and incident cancers**. We first investigated whether the low-frequency variants identified in the current study were associated with mortalities. Only for cancer mortality we observed that the minor G allele of rs79617270 at TERF1 region, which was associated with increased telomere length, was associated with increased cancer mortality [$HR_{95\%CI} =$ 1.544 (1.173, 2.032), Cox regression $P_{Adj} = 0.018$] (Table 2).

The identified low-frequency variants were not associated with CAD in our SCHS CAD data (Supplementary Table 18). We further evaluated if the three low-frequency variants were associated with incident cancers. Significant positive association was observed between the minor G allele of rs79617270 and colorectal cancer [$HR_{95\%CI} = 2.111$ (1.308, 3.406), Cox regression $P_{Adj} = 0.016$]. When analyzing colon and rectal cancer separately, rs79617270 was significantly associated with colon cancer [$HR_{95\%CI} = 2.583$ (1.494, 4.467), Cox regression $P_{Adj} = 0.008$] but not rectal cancer (Table 3). To determine the specific contribution of LTL on the association between rs7961720 and colon cancer risks, mediation analysis was employed. Although significant direct effects between the variant and colon cancer was observed, a significant proportion (4.76%, indirect effect P = 0.041) of the incident cancer effect was determined to be mediated through LTL (Fig. 2 and Supplementary Table 19).

The combined wGRS for increased LTL was significantly associated with increased risk of incident lung cancer [$HR_{95\%CI} = 1.074$ (1.024, 1.125), Cox regression $P_{Adj} = 0.021$] (Table 4). Additionally, we found that the wGRS was strongly associated with lung adenocarcinoma [$HR_{95\%CI} = 1.123$ (1.051, 1.201), Cox regression $P_{Adj} = 0.007$, Table 4] but not with risk of

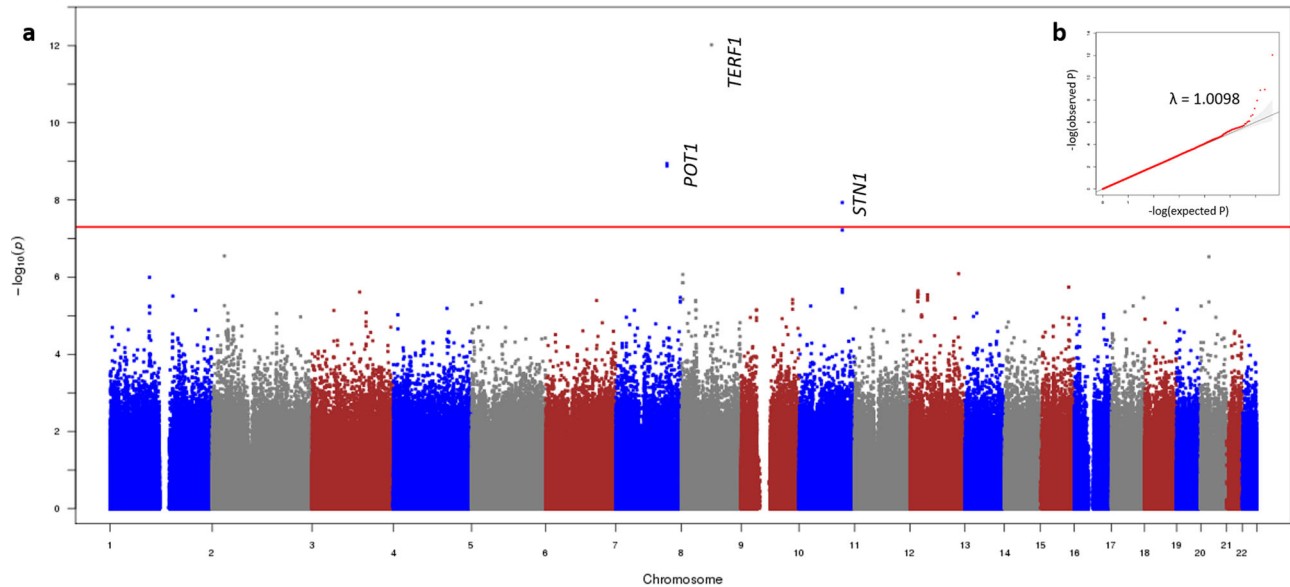

**Fig. 1 Low-frequency LTL associations in the SCHS Discovery dataset (N = 20,177). a** Three loci at chromosome 7, 8, and 10 were identified beyond the genome-wide significance threshold (score test $P < 3.117 \times 10^{-8}$, red line). **b** QQ-plot of observed compared to expected P-values indicated minimal inflation of study results ($\lambda = 1.0098$).

non-adenocarcinoma. The proportion of the effect of wGRS on lung adenocarcinoma mediated through LTL was 24.92% ($P = 1.14 \times 10^{-4}$, Supplemental Table 19 and Fig. 2).

## Discussion

Incidence and mortality of cancers such as lung and colorectal cancers are on the rise in Asia and may display ethnicity-specific genomic disparities[27,28]. Telomere length dysregulation is a central hallmark in carcinogenesis. Our study underscores the value of utilizing non-European ethnic populations to identify low-frequency genetic associations with LTL homeostasis that may also predispose to cancer risks.

POT1 is a well-characterized nuclear protein involved in telomere maintenance[29]. The identified low-frequency missense variant at POT1 was observed to be independent of the previous common index SNP at this locus[4], suggesting a complex genetic control of LTL that may involve both rare and common variants at POT1. Effect estimates at the East Asian-specific low-frequency variant was also 5.3-fold higher than the common index variant, suggesting the importance of low-frequency variants to the genetic predisposition of LTL. Many POT1 mutants that result in the loss of DNA-binding capacity but retain TPP1 binding may lead to an increase of telomere length in cells that have telomerase activity and correspondingly many rare functional variants in POT1 have been associated with cancer risks and longer but more fragile telomeres that promote genomic instability[30–33]. Unlike previous cancer causing POT1 variants, the variant identified in the study was associated with shorter LTL and was not associated with the major cancers evaluated and listed as benign/likely benign in ClinVar (https://www.ncbi.nlm.nih.gov/clinvar/variation/436388/). The putative mutant protein containing the His amino acid at p.410, also did not impact POT1 single-stranded telomeric DNA-binding domains, although the ability to bind TPP1 may be affected.

At TERF1 and STN1 loci, identified low-frequency variants were linked to common index SNPs that we had reported to be specific to East Asians[13]. Given that the identified variants exhibited stronger effect estimates than common SNPs and lie in potentially functional gene elements that exhibit chromatin interactions, these may represent underlying functional variants

at these East Asian-specific LTL loci. Given their strong relevance to telomere length, TERF1 at chromosome 8 and STN1 at chromosome 10 perhaps represent the most likely candidate genes[34–36]. However, compared to functional mutations in gene coding regions, pathogenic regulatory mutations may be more difficult to characterize. Rare variants may significantly affect gene expressions however, there is paucity of data and studies have primarily limited evaluations to rare variants at promoter elements[37]. Our evaluations highlighted chromatin loops that connect regulatory elements containing the identified variants to potential target genes. Especially at the chromosome 8 locus, where chromatin interaction maps from relevant blood cell line was available, it was observed that both TERF1 and SBSPON expression may be influenced. Confirming the specific pathogenic effects of the identified regulatory variants and ascertaining the precise functional genes at these two loci would require substantial functional studies that incorporate evaluations of these ethnicity-specific mutations on the expression and function of neighboring genes, as well as telomere length attrition, at tissue and cell-type-specific resolutions.

Longer LTL has been associated with increased risk of colorectal cancer, including in the Singapore Chinese population[38]. Increased colon telomere length has also been associated with colorectal cancer risk[39] and telomere length of colonic mucosa has been shown to correlate with that of blood leukocytes[40]. In the current study, we observed a consistent association of the minor allele G of rs79617270, which was associated with longer LTL, with increased colon cancer risk. Although it is likely that there were substantial pleiotropic effects between this variant and colon cancer risks, our mediation framework indicated that the specific contribution of cancer risks through LTL pathways was also significant (4.76%). Hence, our data highlights a mechanistic role near the TERF1 gene locus with telomere length and incident colon cancer risk, specific to the East Asian population.

Overall our data indicates that genetically determined LTL was associated with incident lung cancer, particularly lung adenocarcinomas, with about 26% of effects of the wGRS mediated through LTL, but not non-adenocarcinomas. These findings are consistent with previous positive correlation between LTL and lung adenocarcinoma[1,9] and suggest differential susceptibility to

**Table 2 Association of LTL-associated low-frequency variants with mortality in the SCHS dataset.**

| | Case/non-case | rs79314063 | | | | rs79617270 | | | | rs139620151 | | | |
|---|---|---|---|---|---|---|---|---|---|---|---|---|---|
| | | Test allele | HR (95% CI) | P | $P_{adj}$ | Test allele | HR (95% CI) | P | $P_{adj}$ | Test allele | HR (95% CI) | P | $P_{adj}$ |
| All cause mortality | 7743/17,807 | G | 0.906 (0.743, 1.105) | 0.329 | 1.000 | G | 1.177 (0.979, 1.415) | 0.083 | 0.747 | A | 1.018 (0.786, 1.318) | 0.891 | 1.000 |
| Cancer mortality | 2617/22,933 | | 1.109 (0.815, 1.510) | 0.511 | 1.000 | | 1.544 (1.173, 2.032) | 0.002 | **0.018** | | 0.805 (0.492, 1.316) | 0.387 | 1.000 |
| Cardiovascular mortality | 2345/23,205 | | 0.524 (0.326, 0.843) | 0.008 | 0.072 | | 0.965 (0.666, 1.397) | 0.849 | 1.000 | | 1.052 (0.661, 1.672) | 0.832 | 1.000 |
| Ischemic heart disease mortality | 1341/23,205 | | 0.582 (0.322, 1.049) | 0.072 | 0.648 | | 0.905 (0.545, 1.501) | 0.699 | 1.000 | | 0.909 (0.472, 1.751) | 0.775 | 1.000 |
| Stroke mortality | 599/23,205 | | 0.489 (0.184, 1.299) | 0.151 | 1.000 | | 0.922 (0.440, 1.934) | 0.831 | 1.000 | | 1.562 (0.741, 3.291) | 0.241 | 1.000 |
| Respiratory mortality | 1624/23,926 | | 0.943 (0.615, 1.446) | 0.787 | 1.000 | | 0.887 (0.551, 1.427) | 0.621 | 1.000 | | 1.454 (0.889, 2.381) | 0.136 | 1.000 |

$P_{adj}$: score test P adjusted for nine tests, HR hazard ratio, CI confidence interval. $P_{adj}$ < 0.05 values are in bold.

**Table 3 Association of LTL low-frequency risk variants with major incident cancers in the SCHS dataset.**

| | Case/non-case | rs79314063 | | | | rs79617270 | | | | rs139620151 | | | |
|---|---|---|---|---|---|---|---|---|---|---|---|---|---|
| | | Test allele | HR (95% CI) | P | $P_{adj}$ | Test allele | HR (95% CI) | P | $P_{adj}$ | Test allele | HR (95% CI) | P | $P_{adj}$ |
| Colorectal cancer | 654/21,531 | G | 0.837 (0.465, 1.507) | 0.554 | 1.000 | G | 2.111 (1.308, 3.406) | 0.002 | **0.016** | A | 0.941 (0.390, 2.269) | 0.893 | 1.000 |
| Colon cancer | 405/21,531 | | 0.811 (0.387, 1.699) | 0.579 | 1.000 | | 2.583 (1.494, 4.467) | 0.001 | **0.008** | | 0.597 (0.149, 2.397) | 0.467 | 1.000 |
| Rectal cancer | 249/21,531 | | 0.880 (0.332, 2.335) | 0.798 | 1.000 | | 1.326 (0.496, 3.548) | 0.574 | 1.000 | | 1.529 (0.490, 4.776) | 0.465 | 1.000 |
| Lung cancer | 496/21,531 | | 0.763 (0.396, 1.470) | 0.420 | 1.000 | | 1.439 (0.717, 2.888) | 0.306 | 1.000 | | 0.848 (0.272, 2.638) | 0.775 | 1.000 |
| Breast cancer | 335/12,112 | | 0.537 (0.272, 1.060) | 0.073 | 0.584 | | 0.379 (0.095, 1.512) | 0.169 | 1.000 | | 0.344 (0.048, 2.450) | 0.287 | 1.000 |

$P_{adj}$: score test P adjusted for eight tests, HR hazard ratio, CI confidence interval. $P_{adj}$ < 0.05 values are in bold.

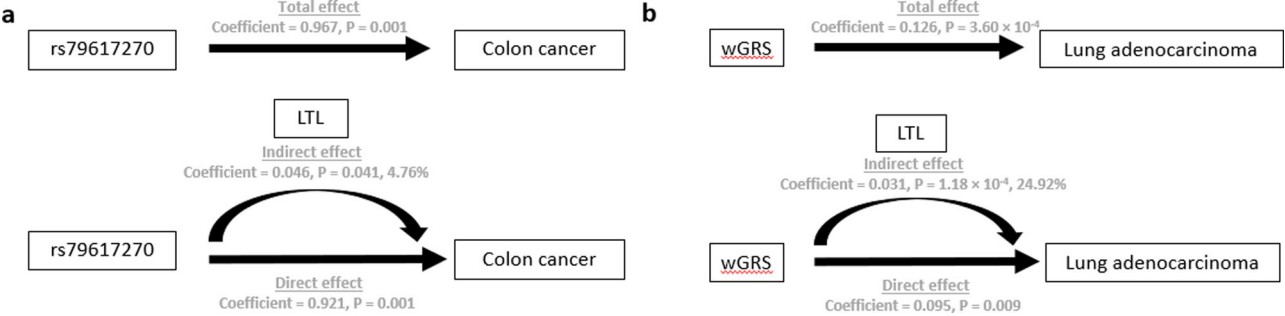

**Fig. 2 Mediation analysis to quantify how much the genetic variant affects incident cancer through LTL. a** Proportion of rs79617270 total effects on colon cancer mediated through LTL = 4.76. **b** Proportion of wGRS total effects on lung adenocarcinoma mediated through LTL = 24.92%.

**Table 4 Association of weighted genetic risk score for increased LTL with major incident cancers in the SCHS dataset.**

|  | Case/non-case | HR (95% CI) | P | $P_{adj}$ |
|---|---|---|---|---|
| Colorectal cancer | 654/21,531 | 1.037 (0.996, 1.081) | 0.079 | 0.553 |
| Colon cancer | 405/21,531 | 1.061 (1.007, 1.117) | 0.025 | 0.175 |
| Rectum cancer | 249/21,531 | 1.001 (0.936, 1.071) | 0.975 | 1.000 |
| Lung cancer | 496/21,531 | 1.074 (1.024, 1.125) | 0.003 | **0.021** |
| Lung adenocarcinoma | 237/21,531 | 1.123 (1.051, 1.201) | 0.001 | **0.007** |
| Non-adenocarcinoma lung cancer | 259/21,531 | 1.031 (0.965, 1.101) | 0.368 | 1.000 |
| Breast cancer | 335/12,112 | 0.999 (0.944, 1.058) | 0.979 | 1.000 |

$P_{adj}$: score test P adjusted for seven tests; HR: hazard ratio; CI: confidence interval. $P_{adj}$ < 0.05 values are in bold.

telomeric dysfunctions in lung cell-types. Inspection at the single SNP level indicated that the lung cancer associations were driven mainly through rs7705526 at the *TERT* gene locus and rs2293607 at the *TERC* gene locus (Supplementary Table 20). The *TERT* locus has been strongly implicated in previous genetic studies for lung cancer[41] and may differentially regulate carcinogenesis of adenocarcinoma as compared to other lung cancers[42]. Taken together, our genetic data strengthens the role of LTL in adenocarcinoma risk and supports the potential utility of LTL as a biomarker for lung adenocarcinoma. At the same time, we note on the discordant lack of association between our LTL wGRS and breast cancer in our study as compared to recent reports indicating that longer LTL was associated with increased risks of breast cancer[43]. It should be noted that the known LTL-associated variants used to generate the wGRS still only explain a modest proportion of phenotypic variance (~4.7%, Supplementary Table 4, as compared to the overall SNP-based heritability of ~20.6% in our study, Supplementary Table 6) and improved genetic instruments may be required to firmly evaluate the role of telomere length in breast cancer.

Limitations in the study should be acknowledged. First, low-frequency variants were generated from imputation procedures rather than direct sequencing and this could have limited the coverage of low-frequency variants. Our imputation procedure, that incorporated local reference panels however, represents a cost-effective approach that has been shown to be effective in identifying disease risk loci[18–20,44]. As we demonstrate the importance of low-frequency variants for LTL and their role in cancer risks, larger scale sequencing approaches would be warranted to uncover the full burden of rare variants in LTL homeostasis. There is also a concern that for the identified *POT1* variant, we were unable to achieve statistically significant association in the replication datasets alone. Study-specific demographic differences and the inclusion of disease cases (CAD and diabetic subjects) in the replication stage of the study may have limited the ability to firmly validate findings from the discovery stage. Nevertheless, all replication datasets, including an independent study utilizing a different experimental platform (WGS), indicated consistent directions of effects and overall meta-analyzed association signals were improved after incorporation of the replication data at all three loci identified in the study.

## Conclusion

In conclusion, our evaluation of East Asian samples identified low-frequency variants that contribute to LTL homeostasis. We further show that LTL-associated variants may affect the risk of developing certain cancers in East Asians.

## Methods

**Study datasets.** The SCHS is a long-term population-based prospective cohort study focused on genetic and environmental determinants of cancer and other chronic diseases in Singapore[45]. Briefly, 63,257 Singaporean Chinese participants consisting of 27,959 men and 35,298 women that were of the two major Chinese dialect groups in Singapore (the Hokkien and the Cantonese) were recruited between April 1993 and December 1998. At recruitment, subjects were interviewed in-person using a structured questionnaire, which included information on demographics, body weight and height, lifetime use of tobacco, alcohol consumption, menstrual/reproductive history (women only), occupational exposure, medical history, and family history of cancer. Three percent random sample of cohort participants were selected for blood and urine sample collection during 1994–1999. At the beginning of year 2000, the collection of blood and urine samples were extended to all surviving members of the entire cohort. During the first follow-up from 1999 to 2004, about half of the cohort participants consented to giving blood or buccal cells for analysis. These studies were approved by the Institutional Review Board at the National University of Singapore and the University of Minnesota, and written informed consent was obtained from all study participants.

The Singapore Study of Macro-angiopathy and Micro-vascular Reactivity in Type 2 Diabetes (SMART2D) dataset is a cross-sectional study conducted between August 2011 and February 2014, including 2057 adults aged 21–90 years with type 2 diabetes mellitus (T2DM). For this study, 969 Singaporean Chinese samples from the SMART2D genotyped on the Illumina HumanOmniExpress-24 Bead Chip were utilized. Diabetic Nephropathy (DN) is an ongoing study conducted from 2002 at the Khoo Teck Puat Hospital, Singapore, which included 1984 adults with T2DM[46]. For this study, 569 Singaporean Chinese samples genotyped on the Illumina HumanOmniZhonghua Bead Chip were utilized. Individual written informed consent was obtained prior to enrollment in these studies and institution's domain-specific ethics approval were obtained.

The SingHEART/Biobank study was established at the National Heart Centre Singapore and is a cohort of normal volunteers enrolled to characterize normal reference values for various cardiovascular and metabolic disease-related markers in Singaporeans[47,48]. Participants were aged between 21 and 69 years without a medical history of myocardial infarction (MI), coronary artery disease (CAD), peripheral arterial disease, stroke, cancer, autoimmune/genetic disease, endocrine disease, diabetes mellitus, psychiatric illness, asthma, or chronic lung disease and chronic infective disease and without a family history of cardiomyopathies[48]. Among study participants, whole genome sequencing (WGS) data for 154 Chinese participants were available for analysis. Written informed consent was obtained for all participants and the SingHealth Centralized Institutional Review Board approved the study.

Basic demographic information for the datasets included in the study is displayed in Supplementary table 1.

**Telomere length measurements.** Telomere length measurements were made from blood leukocytes in the present study. In the SCHS, genomic DNA was extracted from peripheral blood using QIAamp 96 DNA Blood kits (Qiagen, Valencia, CA). A validated monochrome multiplex quantitative PCR (qPCR) method was used to measure LTL[49], which was determined as the ratio of telomere repeat copy number (T) to single (albumin) gene copy number (S) in genomic DNA of study samples relative to a reference sample. 77 samples in the SCHS were used to construct the standard curve for the study. Individual LTL measurements of these samples were within 10% of the cohort mean. The pooled DNA of these 77 participants were run on all qPCR plates, of which 8 replications were performed for each of 4 concentrations: 4, 0.8, 0.16, and 0.032 ng/μL. Thermal cycling was conducted using Applied Biosystem 7900 HT with thermal cycling profile—Stage 1: 15 min at 95 °C; Stage 2: 2 cycles of 15 s at 94 °C, 15 s at 49 °C; and Stage 3: 32 cycles of 15 s at 94 °C, 10 s at 62 °C, 15 s at 74 °C with signal acquisition, 10 s at 84 °C and 15 sec at 88 °C with signal acquisition[4,49]. Telomere length was calculated using real-time PCR cycle thresholds with normalization of telomere length for each 384-well plate. The reproducibility rate of telomere length was excellent, at 3.5%, for all technical sample duplicates. The mean of the two test values was used for statistical analysis.

A similar quantitative qPCR methodology was utilized to measure LTL from peripheral blood DNA from the SMART2D and DN datasets[50]. In these studies, β-globin (S) was used as an internal control to normalize the amount of DNA loaded. For standard curve generation, reference DNA sample (commercially available human G304A DNA) was serially diluted in water by twofold dilution to produce six DNA concentrations from 40 to 1.25 ng in 5 ml. To reduce inter-assay variability, telomere (T) and β-globin (S) were analyzed on the same plate. Additionally, HepG2 DNA (experimental control) was run on each PCR plate, and the average normalizing factor was used to correct the study DNA samples to obtain the final $T/S$ ratio (adjusted $T/S$ ratio). All samples, standards, and controls were run in duplicate and the average values were used for subsequent analyses. The amount of telomeric DNA (T) was divided by the amount of β-globin DNA (S), producing a relative measurement of the LTL ($T/S$ ratio). The coefficients of variation within duplicates of the LTL and β-globin assay were <2%.

In SingHEART/Biobank, WGS data was utilized to estimate LTL[48]. The Telomerecat[51] program was utilized to calculate the ratio between read-pairs mapping to the telomere with reads spanning the telomere boundary and provide base-pair length predictions. The bam2telbam command was run on individual BAM files to generate telbam files that contained sequencing reads relevant to LTL estimation. The telbam2length command was subsequently utilized to generate LTL estimates for the entire cohort's telbam files.

**Mortality and incident cancers.** In the SCHS dataset, all-cause, cardiovascular, respiratory, and cancer deaths from the date of the baseline interview through 31 December 2018 were identified through linkage with the nationwide registry of births and deaths in Singapore. The International Classification of Diseases (ICD) codes used to classify causes of deaths were cardiovascular diseases [ICD9 (390–459) or ICD10 (I00–I99)], respiratory diseases [ICD9 (480–488) or ICD10 (J09-J18)] and cancer [ICD9 (140–208) or ICD10 (C00–C97)]. CAD associations were evaluated in the SCHS CAD dataset[52–55]. Incident cancer cases were identified from the Nationwide Singapore Cancer Registry through annual record linkage analysis for all surviving SCHS study participants. To date, only 47 (<1%) of the total 63,257 original SCHS study participants were known to be lost to follow-up due to migration out of Singapore, suggesting that the ascertainment of cancer incidences among the cohort participants was virtually complete. For this study, four major cancers (colon, rectal, breast, and lung cancers) in the SCHS study with at least 200 incident cases during follow-up and with overlapping genotyping and LTL data were evaluated. Cases of colorectal cancer were determined by the ICD codes C18 for colon cancer and C19-20 for rectal cancer[38]. Breast cancer was defined as the tenth revision of the ICD and Related Health Problems codes C50.0–C50.9[43]. Lung cancer was based on C34 of the International Classifications of Diseases, ICD-O-3[56,57].

**Genotyping and imputation.** Twenty-seven thousand three-hundred eight SCHS samples were genotyped on the Illumina Global Screening Array (GSA). Twenty-five thousand two-hundred seventy-three of these subjects genotyped in year 2018 were utilized as the discovery dataset ("SCHS Discovery") in this study and 2035 additional samples genotyped in year 2020 were utilized in the replication stage ("SCHS Replication"). Quality control (QC) procedures of samples are detailed in Supplementary table 2. Briefly, samples with call-rate < 95.0% and extremes in heterozygozity (> or < 3SD) were excluded. Identity-by-state measures were performed by pair-wise comparison of samples to detect cryptic related samples and one sample, with the lower call-rate, from each relationship was excluded from further analysis. Principal component analysis (PCA) together with 1000 Genomes Projects reference populations and within the SCHS samples were performed to identify possible outliers from reported ethnicity[4].

Additional replication data were drawn from 1,928 independent subjects ("SCHS CAD") from the SCHS CAD-nested case–control study with relevant LTL data[52–55]. LTL data was measured in the SCHS CAD from baseline DNA collected prior to CAD events. GWAS genotyping and QC procedures for this dataset has been detailed[52–55]. GWAS genotyping and QC procedures for the SMART2D ($N$ = 969) and DN ($N$ = 619) type 2 diabetic replication datasets have been described[4,46].

Single-nucleotide polymorphism (SNP) QC are detailed in (Supplementary table 3). We imputed for additional autosomal SNPs using a two reference panel imputation approach by including both the cosmopolitan 1000 Genomes reference panels (Phase 3, 2504 reference panels)[24] and 4810 local Singapore Chinese ($N$ = 2780), Malays ($N$ = 903), and Indians ($N$ = 1127) reference panels from a WGS study of the Singapore populations[22]. Alleles for all SNPs were coded to the forward strand and mapped to HG19. IMPUTE v2[58] was used to mutually impute variants specific to 1000 Genomes and local Singaporean population sample reference panels into each other to obtain a merged reference panel for imputing untyped variants in study datasets. Imputation was performed in chunks of 1 Mb with a buffer of 500 kb and the effective size of the reference population, Ne, was set as 20,000 as recommended[58]. Frequencies of imputed common SNPs (MAF > 1%) showed strong correlation as compared to 1000 Genomes East Asian reference populations (EAS panels, $R$ = 99.7%, Supplementary Fig. 1). For low-frequency variants, we limited analysis to higher quality variants with high imputation confidence scores (info score > 0.8) and those with >20 allele counts (MAF > 0.05% in the SCHS Discovery dataset). This provided 4,598,599 high-quality low-frequency variants with MAF between 0.05% and 1% in the SCHS Discovery set, including 1,915,154 Asian-specific low-frequency variants that were unique to the local Singapore reference panels.

**Statistical analysis.** Twenty-thousand one-hundred seventy-seven participants with genotype and LTL measurements were available for statistical analysis from the SCHS Discovery dataset (Supplementary table 2). For common SNPs (MAF > 1%), standard linear regression analysis was performed[4]. For low-frequency variants (MAF between 0.05% and 1%) Wald's and score tests analyses were evaluated using the rare variant test software (RVTESTS version:20150104)[59]. All analyses were adjusted for age, sex and first three principal components of population stratification. 1,604,004 tagging variants were determined among low-frequency variants analyzed and we utilized a stringent $P < 3.117 \times 10^{-8}$ cutoff to determine genome-wide levels of statistical significance. Low-frequency variants detected at genome-wide levels in the discovery analysis were followed up in replication datasets. Analysis for these variants in each replication dataset were performed as above. Annotations of low-frequency variants were performed using ANNOVAR ver20191024[60] and gene burden tests were performed using RVTESTS (version:20150104)[59]. SNP-based heritability (h²) was evaluated using 2863 samples with 1st degree relationship identified in the SCHS discovery with GCTA-GREML[61,62]. To understand independence of one variant from another, conditional probability analysis was performed where LTL association of one variant was adjusted for the genotypes of the second variant (and vice versa). The generalized linear model (GLM) package in R, adjusted for age and sex was performed using WGS-derived LTL values to evaluate associations of identified variants in the SingHEART/Biobank study.

Association between LTL low-frequency variants and mortalities as well as incident cancers (prevalent cancer cases were excluded from the analysis) were evaluated in the extended SCHS dataset. Subsequently, a weighted genetic risk score (wGRS) was generated by including all low frequency and known common variants reported to be associated with LTL to date[11,14] ($N$ = 21, Supplementary Table 4). At two loci where low-frequency variants were linked to previously reported common index SNPs, only the low-frequency variants were included in the wGRS. The weights for each individual SNP were obtained from meta-analyzed data from previous GWAS for LTL and the current study. It should be noted that the weights for SNPs identified in East Asian samples (common SNPs reported in Dorajoo et al.[4], and low-frequency SNPs from the current study) utilized to generate the LTL wGRS, were obtained from the same SCHS study samples they were originally identified in (Supplementary Table 4). Cox proportional hazards regression, adjusted for age, sex and smoking status (never smokers or ex-smokers vs current), was used to assess the association of individual low-frequency variants and wGRS with mortality/incident cancers. Person-years were calculated for each study participant from the date of the blood drawn to the date of mortality and cancer diagnosis, migration out of Singapore, or 31st December 2015, whichever occurred first. Mediation analysis to quantify how much an independent variable

(genetic variant) affects a dependent variable (incident cancer) through a mediator (LTL) was conducted using the generalized Structural Equation Modeling (gSEM) module in STATA (version15)[63].

**Annotation of identified variants**. Functional annotations for identified variants was performed using the SNP2GENE function in Functional Mapping and Annotation (FUMA v1.3.6)[64]. Datasets used were based on the HG19 human assembly. MAF and $r^2$ were pre-computed using PLINK and based on the EAS 1000 Genomes reference population in FUMA. Potential functionality of variants was evaluated using Combined Annotation Dependent Depletion (CADD)[65], RegulomeDB[66] and core 15-state model of chromatin[67–69]. Hi-C data in FUMA was based on 14 tissue types and seven cell lines from GSE8711211[70]. Chromatin interaction mapping was analyzed as one end, containing identified variants, (region 1) that interacted with the 2nd region of the significant chromatin interactions that mapped to genes whose promoter regions (250 bp up- and 500 bp down-stream of transcriptional start site by default) overlapped with the 2nd end (region 2). Only significant interactions (FDR < 0.05) with mapped genes were retained and for identified chromatin interactions, we reanalyzed genomic regions and plotted regional diagrams using cell line-specific data in FUMA.

**In silico protein predictions**. Predicted tolerance to independent amino acid substitution at p.410 of POT1 was made using SNAP2 software (implemented in PredictProtein[25]), which assesses the potential functional impact of the variant. Solvent accessibility of residues at the POT1 oligonucleotide/oligosaccharide-binding fold (OB-fold) (residues p.146–p.152), stacking residues T1, T2, A3, G4, G5, G6, T7, T8, A9, and G10 that interact with the single-stranded telomeric DNA sequence (residues p.62, 89, 31, 271, 161, 245, 266, and 233), as well as the residues indicated to interact with TPP1 (p.330–634) were predicted using PROFAcc algorithm[71] implemented in ProteinPredict[25].

**Reporting summary**. Further information on research design is available in the Nature Research Reporting Summary linked to this article.

## Data availability

All 4,598,599 variants tested for Singapore Chinese LTL rare variants GWAS in the SCHS Discovery is available in https://figshare.com/articles/dataset/telomere_rare_variants_SCHS_txt/12951689 (https://doi.org/10.6084/m9.figshare.12951689.v1).

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

## Acknowledgements

The Singapore Chinese Health Study was supported by grants from the National Medical Research Council, Singapore (NMRC/CIRG/1456/2016), and the National Institutes of Health (R01 CA144034 and UM1 CA182876). The Singapore Study of Macro-angiopathy and Micro-vascular Reactivity in Type 2 Diabetes (SMART2D) cohort was supported by grants from the National Medical Research Council, Singapore (NMRC/PPG/AH(KTPH)/2011 and NMRC/CIRG/1398/2014). The Diabetic Nephropathy (DN) cohort was supported by grants from Alexandra Health Fund Private Limited (SIG II/15205). Telomere studies in the SMART2D and DN cohorts were supported by Alexandra Health Enabling Grants (AHEG1622 and AHEG1714). The SingHEART/Biobank was supported by the National Medical Research Council, Singapore (NMRC/CG/M006/2017_NHCS). We thank the Singapore Cancer Registry for identification of cancer cases within the Singapore Chinese Health Study. We acknowledge the efforts of all Clinical research coordinators and study participants in the study datasets. S.-C. Lim and W.-P. Koh were supported by National Medical Research Council, Singapore (NMRC/CSA-INV/0020/2017 and NMRC/CSA/0055/2013, respectively). C.C. Khor was supported by National Research Foundation Singapore (NRF-NRFI2018-01).

## Author contributions

R.D., C.-K.H., W.P.K., J.-M.Y., and C.C.K. contributed to the study design. S.L., S.C.L., J.-M.Y., W.P.K., W.K.L., S.D., P.T., and K.K.Y. contributed to the recruitment, sample collection, and data processing. K.B.B., J.A.-H., R.W., W.K.L., Y.M., and R.L.G. generated telomere length data. R.D., Z.L., L.W., W.Y.M., K.S.S., and C.C.K. generated genotyping data. R.D., X.C., R.L.G., L.W., A.J., J.X.T., Y.F., J.L., R.M.v.D., C.C.K., and C.-K.H. contributed to the statistical and bioinformatics analyses. R.D., X.C., C.-K.H. verified underlying data. R.D., X.C., C.-K.H., W.P.K., J.-M.Y., and C.C.K. drafted the manuscript. All authors critically reviewed the manuscript.

## Competing interests

The authors declare no competing interests. C.C.K. is an Editorial Board Member for *Communications Biology*, but was not involved in the editorial review of, nor the decision to publish this article.
