## [Peer Review File · Communications Biology]

Reviewers' comments:

Reviewer #1 (Remarks to the Author):

Chang, et al report results from a GWAS of >20k Singapore Chinese subjects, with replication in ~5k independent subjects. they specifically look for associations between low-frequency variants (MAF: 0.0005-0.01) and leukocyte telomere length. The authors identify 3 independent low-frequency variants associated with inter-individual variation in LTL and explore their association with incident colon, rectal, lung and breast cancers. Overall, this is a methodologically challenging set of analyses that appear to be quite well-conducted. The genomic inflation factor is negligible, likely due in large part to restricting imputed SNPs to those with quite high INFO scores (>0.80). The results identify three rare variants associated with LTL, all in known telomere-regulating genes including POT1, TERF1, and OBFC1. The POT1 result is somewhat confusing, as they identify a deleterious missense SNP that appears to be associated with shorter telomere length. This is in contrast to previous studies of POT1, wherein deleterious SNPs are associated with longer but more fragile telomeres due to shelterin complex dysfunction. Additionally, these loss-of-function variants usually predispose to increased cancer risk (melanoma, lung, CRC, Glioma) due to the associated increase in telomere length and fragility. Some comments:

- 1) As just mentioned, the POT1 results is a little strange relative to other literature. If the minor allele is associated with LoF, then the expectation would be that the minor allele would be associated with longer LTL. Table 1 does not support this. Given that this is a G/C variant, please first just check that you have your reference allele coded properly. If indeed the rare allele is associated with shorter LTL, then this requires some explanation. Is there a chance that this is a hypermorphic allele? Is the common allele actually the hypomorphic allele? The mutation is in the TPP1-binding site, so it would be worth modeling the protein-binding domain and how it may affect binding to ACD.
- 2) The choice to look at associations with only cancer is puzzling, as there are relatively few cancer diagnoses in your cohort and fewer than 500 for any given histology/site group. You've ostensibly found a rare allele in TERF1 that increases telomere length and colon cancer risk. But for the POT1 variant, if it is associated with shortened LTL, then it would be valuable to evaluate this for association with CAD. Indeed, most of your cohorts have cardiovascular health data available, and this looks clearly like a rare variant that would be associated with increased stroke/MI.
- 3) Why are colon and rectal cancer separated? Prior studies typically include them as colorectal cancers, including studies of Shelterin mutations (<https://www.ncbi.nlm.nih.gov/pmc/articles/PMC4917884/>).
- 4) IBS analyses identified >2000 related pairs. Using these related pairs, what is the estimated heritability of LTL in your sample? Does modeling these low-frequency variants substantially improve heritability estimates? This seems like a critical issue of the paper, focusing on low-frequency LTL alleles.
- 5) Why was smoking not adjusted for, where possible? What was the rate of smoking in the samples?
- 6) Please include a supplemental table with the weights used for constructing the wGRS. Ostensibly beta values were used, but should be listed.
- 7) Under the "replication" section in results, please include information for POT1 variant p.V326A, which was recently observed to be a rare lung cancer risk locus in Japanese individuals (PMID: 32514122). It does not appear in your supplemental data and if this is because of INFO score or MAF filters, it would be worth noting.
- 8) Please be specific about direction of association in the results, especially for POT1, i.e. "Rs79314063 was a genotyped missense variant that displayed strong deleterious functional effects (Supplementary table 5) and strong CADD annotation score (scaled CADD = 22.5, top 1% of deleterious variants in the human genome). [The minor (G) allele was associated with significantly shorter LTL (P=xxx)]".
- 9) Make note somewhere that STN1 is also known as OBFC1, which is the more common name in the GWAS literature (ostensibly because that is what UCSC browser uses).
- 10) Reference 53 could probably be better replaced by a reference looking at multiple cancer types, rather than just melanoma. For instance, <https://pubmed.ncbi.nlm.nih.gov/31937561/>

Reviewer #2 (Remarks to the Author):

In this manuscript, the authors presented a genetic study for LTL associations in the East Asian Singapore Chinese population focused on low frequency variants (MAF 0.05%-1%), ascertained through imputation procedures that combined local Singapore population specific reference panels with the 1000 Genomes reference panels. The authors found East Asian specific low frequency variants in or near POT1, TERF1 and STN1 genes were associated with LTL. Mediation analyses showed telomere length partially mediated genetic effects on cancer risk. This study was well done. The analysis was appropriate. The results were interesting and biologically plausible. The manuscript was well written. I only have a couple of minor suggestions:

1. The authors constructed a wGRS to test the association of genetically predicted telomere length with cancer risk. Please add a table to show what SNPs were included, individuals beta values, and what percentage of telomere length these variants explain.
2. In the abstract, the authors stated that "We identified a mechanistic pathway linking the TERF1 rs79617270 variant, LTL and incident colon cancer risks in East Asians". This was an overstatement and somewhat misleading. This was mediation analysis, not mechanistic study. Perhaps simply state that 4.76% of the association between rs79617270 and colon cancer risk was mediated through LTL levels is more proper.

Reviewer #3 (Remarks to the Author):

Dorajoo and colleagues describe their efforts to identify rare genetic variants associated with leukocyte telomere length (LTL) and evaluate whether newly and previously identified genetic variants associated with LTL are also associated with risk for selected cancers in the Singapore Chinese population. Based on imputed genome-wide data (about 4.5 million rare variants), they discovered four novel East Asian-specific rare variants associated with LTL on chromosome 7 (POT1 rs79314063 and rs1896524197), chromosome 8 (TERF1 rs79617270), and chromosome 10 (STN1 rs139620151) in the Singapore Chinese Health Study (SCHS) population (n=20,177) and replicated these associations in four independent data sets (drawn from SCHS, SMART2D, and DN study populations, n=5,356). In the SCHS population (n=21,531), rs79617270 was the only rare variant associated with colon cancer risk. Increased risk of lung cancer, particularly for adenocarcinoma, was also associated with a weighted genetic risk score (wGRS) constructed of novel and known LTL-associated variants. Mediation analyses suggested that a significant proportion of this genetic association was mediated through LTL. This work demonstrates the value of using non-European populations to identify population-specific rare variants associated with LTL homeostasis that may influence predisposition for cancer and thereby advance mechanistic understanding of telomere dysfunction in carcinogenesis. Overall, the general approach appears sound, although certain details about the methods used are either somewhat lacking or not entirely clear. Further discussion on how this work relates to prior studies would also strengthen this manuscript.

Major comments:

1. Comparability of individuals in the discovery and replication phases. Although all individuals are of Singapore Chinese ethnicity, the extent to which individuals included in the discovery and replication phases are similar demographically is not described (e.g., age, sex, etc.). Hence, it is unknown whether the lack of strong evidence for replication may be due in part to population differences. Also, individuals examined in the replication phase include those diagnosed with type 2 diabetes and diabetic nephropathy. It is unclear whether the authors considered whether these or other health conditions might have influenced LTL measurement and subsequently the ability to replicate rare variant associations with LTL.
2. Justification for cancers examined. The rationale for why only four cancers were examined in relation to the novel and previously identified LTL-associated variants should be stated. Presumably, associations with other cancers could have been examined.
3. Details of the population in which genetic associations with cancer risk were examined. Such

associations were evaluated in the “extended SCHS dataset” comprised of 21,531 individuals. It is unclear who was included, as it seems the total number of SCHS participants included in the discovery and replication stages is slightly larger. Providing information on the inclusion/exclusion criteria used would be informative.

4. Methods used to construct wGRS. Additional details on how the wGRS was constructed are needed, including the total number of variants, which variants were selected, and how the weights were obtained. Without this information, others will not be able to replicate construction of this wGRS and apply it in other studies of East Asian populations. Supplementary Table 15 only appears to present previously known common variants. Ideally, weights are obtained from an independent study population in wGRS construction. In this study, the weights appear to come from prior work using the exact same (SCHS) population (ref 11). If so, this caveat should be explicitly acknowledged.

5. Replication of LTL associated low frequency variants identified in European populations. It would be helpful to provide more context on why these analyses were conducted and whether and how they connect to the genetic association analyses with cancer risk that were performed.

6. Interpretation of results from the WGS data. It is stated that “the variants showed similar consistent effects” (line 320). That is a bit confusing, since the scale used to examine LTL in this analysis seems to differ. Presumably what is meant by consistency is in direction (vs. magnitude) of the associations (as stated later on line 409).

7. Balance in discussing findings in the context of prior studies. In the Discussion, the second paragraph notes that prior studies have identified rare functional variants in POT1 that are associated with LTL and cancer risk. However, there is no mention that the identified rare POT1 variant, while associated with LTL, was not associated with risk for any of the four cancers examined. In addition, recently published studies have examined the association between LTL and risk for specific cancers in the SCHS population. One of these studies (ref 29) is referenced, since the results support an association between longer LTL and increased colorectal cancer risk, which is aligned with findings from the present study. The lack of an observed wGRS association with breast cancer risk seems to suggest no clear relationship between LTL and breast cancer risk in Singapore Chinese. However, a prior study found an association between longer LTL and increased breast cancer risk in the SCHS population (ref. 30). These seemingly discordant findings should be discussed, especially since they are derived from the same study population.

8. Interpretation of wGRS association with lung cancer. Results in Supplementary Table 15 suggest that the association is also driven by variation at the TERC locus, along with that at the TERT locus.

9. Overstated conclusions. In this study, only one low frequency variant was associated with colon cancer risk. Therefore, it seems to be a huge stretch to conclude the following in the abstract: “Ethnic-specific and potentially pathogenic coding and non-coding regulatory low frequency variants contribute substantially to LTL homeostasis in East Asians.”

Minor comments:

1. For consistency and clarity, the wording “LTL-associated variants” over “LTL risk variants” is preferable.

2. The authors refer to “LTL levels” in describing certain results. However, the methods do not define any levels. Please clarify.

3. Suggest replacing the label “controls” to “non-cases” in Table 2, since Cox (and not logistic) regression analyses were performed.

4. Suggest presenting the results of associations by histology for lung cancer in the main tables (i.e., Table 3), instead of a supplementary table.

Low frequency variants associated with leukocyte telomere length in the Singapore Chinese population and their role in incident cancers.

Thank you very much for the opportunity to revise our paper. Please find enclosed the revised version of the above referenced manuscript. We are grateful for the reviewers' constructive comments and suggestions. These have been carefully taken into consideration and incorporated into the manuscript accordingly. The revised portions have been highlighted in the revised manuscript for your easy reference. The following paragraphs are our point-by-point reply to these comments.

Reviewers' comments:

Reviewer #1 (Remarks to the Author):

Chang, et al report results from a GWAS of >20k Singapore Chinese subjects, with replication in ~5k independent subjects. they specifically look for associations between low-frequency variants (MAF: 0.0005-0.01) and leukocyte telomere length. The authors identify 3 independent low-frequency variants associated with inter-individual variation in LTL and explore their association with incident colon, rectal, lung and breast cancers. Overall, this is a methodologically challenging set of analyses that appear to be quite well-conducted. The genomic inflation factor is negligible, likely due in large part to restricting imputed SNPs to those with quite high INFO scores (>0.80). The results identify three rare variants associated with LTL, all in known telomere-regulating genes including POT1, TERF1, and OBFC1. The POT1 result is somewhat confusing, as they identify a deleterious missense SNP that appears to be associated with shorter telomere length. This is in contrast to previous studies of POT1, wherein deleterious SNPs are associated with longer but more fragile telomeres due to shelterin complex dysfunction. Additionally, these loss-of-function variants usually predispose to increased cancer risk (melanoma, lung, CRC, Glioma) due to the associated increase in telomere length and fragility. Some comments:

- 1) As just mentioned, the POT1 results is a little strange relative to other literature. If the minor allele is associated with LoF, then the expectation would be that the minor allele would be associated with longer LTL. Table 1 does not support this. Given that this is a G/C variant, please first just check that you have your reference allele coded properly. If indeed the rare allele is associated with shorter LTL, then this requires some explanation. Is there a chance that this is a hypermorphic allele? Is the common allele actually the hypomorphic allele? The mutation is in the TPP1-binding site, so it would be worth modeling the protein-binding domain and how it may affect binding to ACD.

Response: We thank the reviewer for clarifying this. We have checked all our analysis files and ensured that the reference allele had been coded correctly. The minor G allele of this variant, rs79314063 (p.Asp410His), has frequency of 0.7% in our study and this is consistent with the EAS 1000 Genomes reference panel (G allele frequency = 0.6%) as well as the gnomAD EAS exome (G allele frequency = 0.6%) and genome (G allele frequency = 1.0%) panels.

The minor G allele of rs79314063 was associated with reduced leukocyte telomere length (LTL) in our study (beta = -0.320, meta P = 6.94×10^{-10}). This variant displayed high CADD score (22.5) and potentially deleterious effects in SIFT, Polyphen2 and Mutation Taster. However, the clinical significance of this variant is listed as “benign/likely benign” in ClinVar (<https://www.ncbi.nlm.nih.gov/clinvar/variation/436388/>).

POT1 mutants that lose DNA binding capacity but retain TPP1 binding may lead to an increase of telomere length in cells that have telomerase activity¹⁻³. As recommended by the reviewer, we have performed additional *in silico* predictions on the putative mutant protein containing the His amino acid at position p.410 and compared results with the wildtype POT1 protein. The potential functional impact of the variant was evaluated using SNAP2⁴ and indicated a relatively high deleterious effect of 73 (85% accuracy) with the change of residue from Asp to His at position p.410. We next evaluated if the putative mutant protein containing the His residue at position p.410 affects the oligonucleotide/oligosaccharide-binding fold (OB-fold) (residues p.146–p.152), stacking residues T1, T2, A3, G4, G5, G6, T7, T8, A9 and G10 that interact with the single-stranded telomeric DNA sequence (residues p.62, 89, 31, 271, 161, 245, 266 and 233) as well as the residues reported to interact with TPP1 (p.330–634) using solvent accessibility

prediction⁴⁻⁶. Unlike most *POT1* variants linked to cancer that are located in the N-terminal OB-fold domains and affect DNA binding capacity, the putative mutant protein containing the His amino acid at p.410 identified in our study, did not affect solvent accessibility of residues at the OB-fold and stacking residues of POT1. However, the His amino acid at p.410 modified solvent accessibility of 17 out of 305 residues within the TPP1 interacting domain as compared to the wildtype protein (p.391, 396, 413, 417, 419, 451, 196, 534 and 575 were changed to exposed while p.336, 399, 418, 440, 553, 592, 504 and 628 were changed to buried with the His residue at p.410) (Supplementary table 9). As such the identified LTL decreasing variant, rs79314063 (p.Asp410His), did not strongly impact single-stranded telomeric DNA binding domains of POT1, although ability to bind TPP1 may be affected.

These additional *in silico* data have been presented in Supplementary table 9 and in the results, methods and discussion sections of the revised manuscript.

“The change from Asp to His residue at p.410 in POT1 indicated a relatively high deleterious effect score of 73 (85% accuracy) using SNAP2⁴. The putative mutant protein containing the His amino acid at p.410, did not affect solvent accessibility of residues at the OB-fold and stacking residues of POT1. However, the His amino acid at p.410 modified solvent accessibility of 17 out of 305 residues within the TPP1 interacting domain as compared to the wildtype protein (p.391, 396, 413, 417, 419, 451, 196, 534 and 575 were changed to exposed while p.336, 399, 418, 440, 553, 592, 504 and 628 were changed to buried with the His residue at p.410) (Supplementary table 9).” (Results, Page 14-15, line 261-268).

“Many POT1 mutants that result in the loss of DNA binding capacity but retain TPP1 binding may lead to an increase of telomere length in cells that have telomerase activity and correspondingly many rare functional variants in *POT1* have been associated with cancer risks and longer but more fragile telomeres that promote genomic instability^{1-3, 7}. Unlike previous cancer causing *POT1* variants, the variant identified in the study was associated with shorter LTL and was not associated with the major cancers evaluated and listed as benign/likely benign in ClinVar (<https://www.ncbi.nlm.nih.gov/clinvar/variation/436388/>). The putative mutant protein containing the His amino acid at p.410, also did not impact POT1 single-stranded telomeric DNA

binding domains, although the ability to bind TPP1 may be affected.” (Discussion, Page 18, line 351-359)

“Predicted tolerance to independent amino acid substitution at p.410 of POT1 was made using SNAP2 software (implemented in PredictProtein⁴), which assesses the potential functional impact of the variant. Solvent accessibility of residues at the POT1 oligonucleotide /oligosaccharide-binding fold (OB-fold) (residues p.146–p.152), stacking residues T1, T2, A3, G4, G5, G6, T7, T8, A9 and G10 that interact with the single-stranded telomeric DNA sequence (residues p.62, 89, 31, 271, 161, 245, 266 and 233) as well as the residues indicated to interact with TPP1 (p.330–634) were predicted using PROFAcc algorithm⁸ implemented in ProteinPredict⁴. (Methods, Page 13, line 229-235)

- 2) The choice to look at associations with only cancer is puzzling, as there are relatively few cancer diagnoses in your cohort and fewer than 500 for any given histology/site group. You've ostensibly found a rare allele in TERF1 that increases telomere length and colon cancer risk. But for the POT1 variant, if it is associated with shortened LTL, then it would be valuable to evaluate this for association with CAD. Indeed, most of your cohorts have cardiovascular health data available, and this looks clearly like a rare variant that would be associated with increased stroke/MI.

Response: We thank the reviewer for this suggestion. Rs79314063 did not show a significant association with coronary artery disease (CAD) association in our SCHS-CAD study dataset (HR = 1.835 (0.820, 4.104), P_{Adj} = 0.420) (Table 1 below).

Table 1: Association of LTL associated low frequency variants with coronary artery disease in the SCHS dataset.

Case/Non-Case = 711 / 1,246												
rs79314063				rs79617270				rs139620151				
	TA	HR (95% CI)	P	P _{adj}	TA	HR (95% CI)	P	P _{adj}	TA	HR (95% CI)	P	P _{adj}
		1.835				0.844				1.098		
CAD	G	(0.820, 4.104)	0.140	0.420	G	(0.350, 2.036)	0.844	1.000	A	(0.546, 2.206)	0.793	1.000

P_{adj}: score test P adjusted for 3 tests; TA: test allele; HR: hazard ratio; CI: confidence interval.

We have also evaluated the identified variants for overall cardiovascular disease mortality as well as ischemic heart disease mortality and stroke mortality, separately. Rs79314063 in *POT1* was not significantly associated with these cardiovascular disease mortalities (Table 2 below). The rare allele of *TERF1* (rs79617270) however, indicated a significant risk with cancer mortality in our data and we additionally evaluated for incident cancer risks (Table 2 below).

Table 2: Association of LTL associated low frequency variants with mortality in the SCHS dataset.

	rs79314063					rs79617270					rs139620151						
	Case/Non-Case	Test allele	HR (95% CI)	P	P _{adj}	Test allele	HR (95% CI)	P	P _{adj}	Test allele	HR (95% CI)	P	P _{adj}	Test allele	HR (95% CI)	P	P _{adj}
All cause mortality	7,743 / 17,807	G	0.906 (0.743, 1.105)	0.329	1.000	G	1.177 (0.979, 1.415)	0.083	0.747	A	1.018 (0.786, 1.318)	0.891	1.000				
Cancer mortality	2,617 / 22,933		1.109 (0.815, 1.510)	0.511	1.000		1.544 (1.173, 2.032)	0.002	0.018		0.805 (0.492, 1.316)	0.387	1.000				
Cardiovascular mortality	2,345 / 23,205		0.524 (0.326, 0.843)	0.008	0.072		0.965 (0.666, 1.397)	0.849	1.000		1.052 (0.661, 1.672)	0.832	1.000				
Ischemic heart disease mortality	1,341 / 23,205		0.582 (0.322, 1.049)	0.072	0.648		0.905 (0.545, 1.501)	0.699	1.000		0.909 (0.472, 1.751)	0.775	1.000				
Stroke mortality	599 / 23,205		0.489 (0.184, 1.299)	0.151	1.000		0.922 (0.440, 1.934)	0.831	1.000		1.562 (0.741, 3.291)	0.241	1.000				
Respiratory mortality	1,624 / 23,926	0.943 (0.615, 1.446)	0.787	1.000	0.887 (0.551, 1.427)	0.621	1.000	1.454 (0.889, 2.381)	0.136	1.000							

P_{adj}: score test P adjusted for 9 tests; HR: hazard ratio; CI: confidence interval.

The additional data have been presented in the revised manuscript.

“Association between LTL low frequency variants and mortality as well as incident cancers (prevalent cancer cases were excluded from the analysis) were evaluated in the extended SCHS dataset.” (Page 11, line 197-198)

“We first investigated whether the low frequency variants identified in the current study were associated with mortality. The minor G allele of rs79617270 at *TERF1* region, which was associated with increased telomere length, was associated with increased cancer mortality [HR_{95%CI}= 1.544 (1.173, 2.032), Cox regression P_{Adj} =0.018] (Table 2).” (Results, Page 16, line 318-321)

“The identified low frequency variants were not associated with CAD in our SCHS CAD data (Supplementary table 18).” (Results, Page 17, line 323-324)

“In the SCHS dataset, all-cause, cardiovascular, respiratory, and cancer deaths from the date of the baseline interview through 31 December 2018 were identified through linkage with the nationwide registry of births and deaths in Singapore. The International Classification of Diseases (ICD) codes for the mortality has been described in detail previously. CAD associations were evaluated in the SCHS CAD dataset⁹⁻¹².” (Methods, Page 8-9, line 130-134).

3) Why are colon and rectal cancer separated? Prior studies typically include them as colorectal cancers, including studies of Shelterin mutations

Response: Mechanisms and pathways involved in colon and rectal carcinogenesis may be varied^{13, 14}. As suggested by the reviewer, we have now also combined and evaluated colon and rectal cancers as colorectal cancers. We have included the association between LTL low frequency risk variants with colorectal cancers in Table 3-4 and made the following changes in the revised manuscript:

“We further evaluated if the three low frequency variants were associated with incident cancers. Significant positive association was observed between the minor G allele of rs79617270 and colorectal cancer [HR_{95%CI}= 2.111 (1.308, 3.406), Cox regression P_{Adj} =0.016].” (Results Page 17, line 324-326)

Table 3: Association of LTL associated low frequency variants with major incident cancers in the SCHS dataset.

	Case/Non-case	rs79314063				rs79617270				rs139620151			
		Test allele	HR (95% CI)	P	P _{adj}	Test allele	HR (95% CI)	P	P _{adj}	Test allele	HR (95% CI)	P	P _{adj}
Colorectal cancer	654 / 21,531	0.837 (0.465, 1.507)	0.554	1.000	G	2.111 (1.308, 3.406)	0.002	0.016	A	0.941 (0.390, 2.269)	0.893	1.000	
Colon cancer	405 / 21,531	0.811 (0.387, 1.699)	0.579	1.000	G	2.583 (1.494, 4.467)	0.001	0.008	A	0.597 (0.149, 2.397)	0.467	1.000	
Rectal cancer	249 / 21,531	0.880 (0.332, 2.335)	0.798	1.000	G	1.326 (0.496, 3.548)	0.574	1.000	A	1.529 (0.490, 4.776)	0.465	1.000	
Lung cancer	496 / 21,531	0.763 (0.396, 1.470)	0.420	1.000	G	1.439 (0.717, 2.888)	0.306	1.000	A	0.848 (0.272, 2.638)	0.775	1.000	
Breast cancer	335 / 12,112	0.537 (0.272, 1.060)	0.073	0.584	G	0.379 (0.095, 1.512)	0.169	1.000	A	0.344 (0.048, 2.450)	0.287	1.000	

P_{adj}: score test P adjusted for 8 tests; HR: hazard ratio; CI: confidence interval.

4) IBS analyses identified >2000 related pairs. Using these related pairs, what is the estimated heritability of LTL in your sample? Does modeling these low-frequency variants substantially improve heritability estimates? This seems like a critical issue of the paper, focusing on low-frequency LTL alleles.

Response: We thank the reviewer for this suggestion. Using 2,863 samples with 1st degree relationship identified in the SCHS discovery dataset we have utilized Genome-wide Complex Trait Analysis (GCTA) to estimate SNP-based heritability (h^2) using genome-wide SNPs (MAF > 0.05%), common SNPs (MAF > 1%) and low-frequency SNPs (MAF between 0.05% and 1%). Modeling low-frequency variants

improved LTL heritability estimates and they explain approximately 2.9%, indicating that such variants may influence LTL levels (Table 4 below). We have included these data in the revised manuscript.

Table 4: SNP based heritability of genome-wide SNPs, common SNPs and low-frequency SNPs among 2,863 1st degree related subjects from the SCHS study.

Variants	SNP-based heritability
Common SNPs (MAF > 1%)	0.187
This study (MAF between 0.05% and 1%)	0.029
Genome-wide SNPs (MAF > 0.05%)	0.206

“Inclusion of low-frequency variants (MAF between 0.05% and 1%) improved LTL heritability estimates in our study and SNP-based h^2 of low-frequency variants was approximately 2.9% in the SCHS discovery dataset (Supplementary table 6).” (Results, Page 13, line 247-249)

“SNP-based heritability (h^2) was evaluated using 2,863 samples with 1st degree relationship identified in the SCHS discovery with GCTA-GREML^{15,16}.” (Methods, Page 11, line 190-191)

5) Why was smoking not adjusted for, where possible? What was the rate of smoking in the samples?

Response: Smoking status for the SCHS study samples were available through questionnaire data. Proportion of never, ex and current smokers were 70.81%, 11.15% and 18.04%. For the identified SNPs in the study we have now further adjusted LTL associations on smoking status (never smokers or ex-smokers vs current). These adjustments did not change the LTL effects of the identified SNPs in the study (Table 5 below). These new data are presented in Supplementary table 15 and in page 16 line 300-301 of the revised manuscript.

“Cox proportional hazards regression, adjusted for age, sex and smoking status (never smokers or ex-smokers vs current), was used to assess the association of individual low frequency variants and wGRS with mortality/incident cancers.” (Methods, Page 12, line 206-209)

“Further adjustment for smoking status did not affect the LTL associations of these SNPs (Supplementary table 15).” (Results, Page 16, line 302-303)

Table 5: Association of identified SNPs with LTL after further adjustment for smoking status (never smokers or ex-smokers vs current)

SNP	Chr	Position	Gene	TA	TAF	SCHS Discovery			SCHS CAD			SCHS Replication			Meta			
						Beta	Se	P	Beta	Se	P	Beta	Se	P	Beta	Se	P	Q _{total}
rs79314063	7	1.24E+08	POT1	G	0.007	-0.345	0.057	1.06 × 10 ⁻⁹	-0.262	0.456	0.566	-0.099	0.169	0.557	-0.319	0.053	1.97 × 10 ⁻⁹	0.384
rs79617270	8	73978144	TERF1/SBSPON	G	0.008	0.384	0.054	9.10 × 10 ⁻¹³	0.494	0.268	0.050	0.326	0.168	0.052	0.383	0.050	2.73 × 10 ⁻¹⁴	0.865
rs139620151	10	1.06E+08	STN1 / SH3PXD2A	A	0.006	0.383	0.067	1.09 × 10 ⁻⁸	0.486	0.257	0.058	0.110	0.431	0.799	0.384	0.064	2.23 × 10 ⁻⁹	0.755

6) Please include a supplemental table with the weights used for constructing the wGRS. Ostensibly beta values were used, but should be listed.

Response: We thank the reviewer for bringing up this important point and have included the related information in Supplementary table 4.

7) Under the "replication" section in results, please include information for POT1 variant p.V326A, which was recently observed to be a rare lung cancer risk locus in Japanese individuals (PMID: 32514122). It does not appear in your supplemental data and if this is because of INFO score or MAF filters, it would be worth noting.

Response: The POT1 variant p.V326A was observed with low INFO score after imputation. INFO scores were 0.548, 0.092 and 0.625 in the SCHS Discovery, SCHS CAD and SCHS replication datasets, respectively. We have now indicated that this variant was excluded from analysis due to the INFO score threshold (> 0.8) used in the study.

"The POT1 p.V326A variant, recently identified in the Japanese population was not tested in the present study as it did not pass imputation info score threshold (info score between 0.094 and 0.625)¹⁷" (Results, Page 16, line 312-313)

- 8) Please be specific about direction of association in the results, especially for POT1, i.e. "Rs79314063 was a genotyped missense variant that displayed strong deleterious functional effects (Supplementary table 5) and strong CADD annotation score (scaled CADD = 22.5, top 1% of deleterious variants in the human genome [The minor (G) allele was associated with significantly shorter LTL (P=xxx)]".

Response: We thank the reviewer for highlighting this. We have accordingly revised the manuscript:

"Rs79314063 was a genotyped missense variant that displayed strong deleterious functional effects (Supplementary table 8) and strong CADD annotation score (scaled CADD = 22.5, top 1% of deleterious variants in the human genome [The minor (G) allele was associated with significantly shorter LTL ($P = 6.94 \times 10^{-10}$)])." (Page 14, line 258-261)

- 9) Make note somewhere that STN1 is also known as OBFC1, which is the more common name in the GWAS literature (ostensibly because that is what UCSC browser uses).

Response: We thank the reviewer for highlighting this. We have indicated that STN1 is also known as OBFC1 on page 15, line 275-276.

"...rs12415148 at the *STN1* locus (also known as *OBFC1*) in chromosome 10..." (Page 15, line 276-277)

- 10) Reference 53 could probably be better replaced by a reference looking at multiple cancer types, rather than just melanoma. For instance, <https://pubmed.ncbi.nlm.nih.gov/31937561/>

Response: As suggested, we have replaced reference 53 (Currently reference 58).

58. Shen E, Xiu J, Lopez GY, et al. POT1 mutation spectrum in tumour types commonly diagnosed among POT1-associated hereditary cancer syndrome families. *J Med Genet.* 2020; 57: 664-70.

Reviewer #2 (Remarks to the Author):

In this manuscript, the authors presented a genetic study for LTL associations in the East Asian Singapore Chinese population focused on low frequency variants (MAF 0.05%-1%), ascertained through imputation procedures that combined local Singapore population specific reference panels with the 1000 Genomes reference panels. The authors found East Asian specific low frequency variants in or near POT1, TERF1 and STN1 genes were associated with LTL. Mediation analyses showed telomere length partially mediated genetic effects on cancer risk. This study was well done. The analysis was appropriate. The results were interesting and biologically plausible. The manuscript was well written. I only have a couple of minor suggestions:

1. The authors constructed a wGRS to test the association of genetically predicted telomere length with cancer risk. Please add a table to show what SNPs were included, individuals beta values, and what percentage of telomere length these variants explain.

Response: We thank the reviewer for bringing up this important point and have included the requested information in Supplementary table 4.

2. In the abstract, the authors stated that “We identified a mechanistic pathway linking the TERF1 rs79617270 variant, LTL and incident colon cancer risks in East Asians”. This was an overstatement and somewhat misleading. This was mediation analysis, not mechanistic study. Perhaps simply state that 4.76% of the association between rs79617270 and colon cancer risk was mediated through LTL levels is more proper.

Response: We thank the reviewer for highlighting this. We have modified the abstract as suggested.

“4.76% of the association between the *TERF1* rs79617270 variant and colon cancer risk was mediated through LTL.” (Abstract, Page 3, line 10-11)

Reviewer #3 (Remarks to the Author):

Dorajoo and colleagues describe their efforts to identify rare genetic variants associated with leukocyte telomere length (LTL) and evaluate whether newly and previously identified genetic variants associated with LTL are also associated with risk for selected cancers in the Singapore Chinese population. Based on imputed genome-wide data (about 4.5 million rare variants), they discovered four novel East Asian-specific rare variants associated with LTL on chromosome 7 (POT1 rs79314063 and rs1896524197), chromosome 8 (TERF1 rs79617270), and chromosome 10 (STN1 rs139620151) in the Singapore Chinese Health Study (SCHS) population (n=20,177) and replicated these associations in four independent data sets (drawn from SCHS, SMART2D, and DN study populations, n=5,356). In the SCHS population (n=21,531), rs79617270 was the only rare variant associated with colon cancer risk. Increased risk of lung cancer, particularly for adenocarcinoma, was also associated with a weighted genetic risk score (wGRS) constructed of novel and known LTL-associated variants. Mediation analyses suggested that a significant proportion of this genetic association was mediated through LTL. This work demonstrates the value of using non-European populations to identify population-specific rare variants associated with LTL homeostasis that may influence predisposition for cancer and thereby advance mechanistic understanding of telomere dysfunction in carcinogenesis. Overall, the general approach appears sound, although certain details about the methods used are either somewhat lacking or not entirely clear. Further discussion on how this work relates to prior studies would also strengthen this manuscript.

Major comments:

1. Comparability of individuals in the discovery and replication phases. Although all individuals are of Singapore Chinese ethnicity, the extent to which individuals included in the discovery and replication phases are similar demographically is not described (e.g., age, sex, etc.). Hence, it is unknown whether the lack of strong evidence for replication may be due in part to population differences. Also, individuals examined in the replication phase include those diagnosed with type 2 diabetes and diabetic nephropathy. It is unclear whether the authors considered whether these or other health conditions might have influenced LTL measurement and subsequently the ability to replicate rare variant associations with LTL.

Response: We thank the reviewer for highlighting this. We have highlighted the related information in table 6 (below). Proportion of males and smoking status were higher among the SCHS CAD dataset as compared to the SCHS population datasets.

Also, given the influence of diabetes with LTL, the diabetic datasets also had lower mean LTL as compared to the SCHS population datasets. As the reviewer correctly pointed out, these differences between datasets used in the study may have affected the ability to firmly replicate findings and we have now indicated this as a limitation on the study. Nevertheless, at the specific variants detected for LTL associations in the current study, direction of effects were consistent in all datasets utilized in the replication stage of the study and improved initial associations observed in the discovery dataset. We have made the following changes to the revised manuscript:

Table 6: Demographic information for the datasets included in the study. Data presented and mean (SD).

Ethnicity	SCHS Discovery	SCHS Replication	SCHS CAD cases	SCHS CAD controls	SMART2D	DN
	N = 20,177	N = 1,840	N = 704	N = 1,224	N = 969	N = 619
Age (years)	55.06 (7.45)	54.99 (7.52)	59.54 (7.92)	59.05 (7.82)	58.68 (11.51)	58.24 (12.25)
Male (%)	44.57	37.94	64.68	62.95	54.75	60.57
Smoking ever (%)	10.92	9.54	14.04	15.80	-	-
current (%)	17.50	14.69	33.76	22.96	-	-
LTL	1.02 (0.22)	1.05 (0.26)	1.00 (0.23)	1.02 (0.24)	0.71 (0.31)	0.85 (0.32)

LTL: leucocyte telomere length

“Basic demographic information for the datasets included in the study is displayed in Supplementary table 1.” (Page 7, Line 94-95)

“Study specific demographic differences and the inclusion of disease cases (CAD and diabetic subjects) in the replication stage of the study may have limited the ability to firmly validate findings from the discovery stage. Nevertheless, all replication datasets, including an independent study utilizing a different experimental platform (WGS), indicated consistent directions of effects and overall meta-analyzed association signal was improved after incorporation of the replication data at all three loci identified in the study.” (Discussion,

2. Justification for cancers examined. The rationale for why only four cancers were examined in relation to the novel and previously identified LTL-associated variants should be stated. Presumably, associations with other cancers could have been examined.

Response: These cancers included in the current study were the major incident cancer types in the SCHS study, with > 200 incident cases that had overlapping LTL and genotyping data. We have now indicated this in the revised version of the manuscript.

“For this study, four major cancers (colon, rectal, breast and lung cancers) in the SCHS study with at least 200 incident cases during follow-up and with overlapping genotyping and LTL data were evaluated.” (Methods, Page 9, line 139-140).

3. Details of the population in which genetic associations with cancer risk were examined. Such associations were evaluated in the “extended SCHS dataset” comprised of 21,531 individuals. It is unclear who was included, as it seems the total number of SCHS participants included in the discovery and replication stages is slightly larger. Providing information on the inclusion/exclusion criteria used would be informative.

Response: We thank the reviewer for highlighting this point. The number of individuals included in the genetic associations with cancer risk was slightly smaller from that in the GWAS study because only incident cancer cases were included and we had excluded the prevalent cases. We have added the following sentence in the revised manuscript to clarify:

“Association between LTL low frequency variants and mortality as well as incident cancers (prevalent cancer cases were excluded from the analysis) were evaluated in the extended SCHS dataset.” (Page 11, line 197-198)

4. Methods used to construct wGRS. Additional details on how the wGRS was constructed are needed, including the total number of variants, which variants were selected, and how the weights were obtained. Without this information, others will not be able to replicate construction of this wGRS and apply it in other studies of East Asian populations. Supplementary Table 15 only appears to present previously known common variants. Ideally, weights are

obtained from an independent study population in wGRS construction. In this study, the weights appear to come from prior work using the exact same (SCHS) population (ref 11). If so, this caveat should be explicitly acknowledged.

Response: We thank the reviewer for bringing up this important point and have included the related information in Supplementary table 4 (Table 7 below) and revised the content accordingly in the manuscript:

Table 7: LTL associated SNPs included in the weighted genetic risk score (wGRS).

	Chr	Position	Gene	Effect size	Variance explained %	Source
rs3219104	1	226562621	PARP1	0.074	0.330	Dorajoo R et al. 2019
rs11890390	2	54485682	ACYP2	0.040	0.130	Dorajoo R et al. 2019
rs55749605	3	101232093	SENP7	0.037	0.010	Li C et al. 2020
rs13137667	4	71774347	MOB1B	0.077	0.030	Dorajoo R et al. 2019
rs2293607	3	169482335	TERC	0.120	0.830	Li C et al. 2020
rs10857352	4	164101482	NAF1	0.064	0.150	Dorajoo R et al. 2019
rs7705526	5	1285974	TERT	0.118	0.800	Dorajoo R et al. 2019
rs2736176	6	31587561	PRRC2A	0.035	0.020	Li C et al. 2020
rs79314063	7	124481168	POT1	0.320	0.170	Current Study
rs7776744	7	124599749	POT1	0.058	0.170	Dorajoo R et al. 2019
rs79617270	8	73978144	TERF1	0.376	0.230	Current Study
rs7095953	10	101274425	NKX2-3	0.047	0.210	Dorajoo R et al. 2019
rs139620151	10	105593428	STN1	0.394	0.130	Current Study
rs227080	11	108247888	ATM	0.060	0.070	Dorajoo R et al. 2019
rs41293836	14	24721327	TINF2	0.233	0.840	Dorajoo R et al. 2019
rs2302588	14	73404752	DCAF4	0.042	0.030	Dorajoo R et al. 2019
rs3785074	16	69406986	TERF2	0.035	0.050	Li C et al. 2020
rs2967374	16	82209861	MPHOSPH6	0.056	0.100	Dorajoo R et al. 2019
rs1001761	18	662103	TYMS	0.042	0.160	Dorajoo R et al. 2019
rs7253490	19	22293706	ZNF257	0.036	0.110	Dorajoo R et al. 2019
rs41309367	20	62309554	RTEL1	0.058	0.150	Dorajoo R et al. 2019

“Subsequently, a weighted genetic risk score (wGRS) was generated by including all novel and known variants reported to be associated with LTL to date^{11, 14} (N = 21, Supplementary table 4). At two loci where novel low frequency variants were linked to previously reported common index SNPs, only the low frequency variants were included in the wGRS. The weights for each individual SNP were obtained from meta-analysed data from previous GWAS for LTL and the

current study. It should be noted that the weights for SNPs identified in East-Asian samples (common SNPs reported in Dorajoo R et al, 2019 and low-frequency SNPs from the current study) utilized to generate the LTL wGRS, were obtained from the same SCHS study samples they were originally identified in (Supplementary table 4).” (Page 11, line 199-206)

5. Replication of LTL associated low frequency variants identified in European populations. It would be helpful to provide more context on why these analyses were conducted and whether and how they connect to the genetic association analyses with cancer risk that were performed.

Response: The van der Spek A et al, 2020 study was the only relatively large-scale study (as far as we were aware) that had reported associations of low frequency variants (at *ATM* and *RPL8*) with leukocyte telomere length (LTL) in the European ancestry population. As such we evaluated if there might be similar LTL associated low frequency variants at these gene loci in our East-Asian data. A low frequency missense variant at *ATM* (rs3218670) was associated with reduced LTL in our study samples. However, given the very nominal significance levels observed (LTL association $P = 0.003$), we did not bring this variant forward for additional cancer and other disease evaluations in the study.

As also suggested by reviewer 1 and given the strong role of rare variants at *POT1* in human cancers, we have now also looked-up the *POT1* p.V326A variant, recently identified in the Japanese population. However, this variant was observed with low imputation info score in our datasets and was thus excluded from our analysis. It should also be noted that our gene burden test only highlighted the *POT1* gene as strongly associated with LTL in our data and this association was primarily driven by the single missense variant, rs79314063, we report in our study.

6. Interpretation of results from the WGS data. It is stated that “the variants showed similar consistent effects” (line 320). That is a bit confusing, since the scale used to examine LTL in this analysis seems to differ. Presumably what is meant by consistency is in direction (vs. magnitude) of the associations (as stated later on line 409).

Response: We thank the reviewer for bringing up this point and have revised the manuscript accordingly:

“The variants showed similarly consistent direction of effects in this dataset and rs139620151 was statistically significant ($P=0.009$, Supplementary table 14).” (Page 16, line 300-302)

7. Balance in discussing findings in the context of prior studies. In the Discussion, the second paragraph notes that prior studies have identified rare functional variants in *POT1* that are associated with LTL and cancer risk. However, there is no mention that the identified rare *POT1* variant, while associated with LTL, was not associated with risk for any of the four cancers examined. In addition, recently published studies have examined the association between LTL and risk for specific cancers in the SCHS population. One of these studies (ref 29) is referenced, since the results support an association between longer LTL and increased colorectal cancer risk, which is aligned with findings from the present study. The lack of an observed wGRS association with breast cancer risk seems to suggest no clear relationship between LTL and breast cancer risk in Singapore Chinese. However, a prior study found an association between longer LTL and increased breast cancer risk in the SCHS population (ref. 30). These seemingly discordant findings should be discussed, especially since they are derived from the same study population.

Response: We agree with the reviewer and have clarified on these points in the revised manuscript. We have now indicated that the *POT1* variant identified in our study was associated with reduced LTL and was not associated with cancers in our study.

“Unlike previous cancer causing *POT1* variants^{5, 6}, the variant identified in the study was associated with shorter LTL and was not associated with the major cancers evaluated and listed as benign/likely benign in ClinVar (<https://www.ncbi.nlm.nih.gov/clinvar/variation/436388/>).” (Discussion, Page 18, line 354-357)

It should be noted that large-scale Mendelian randomization studies using LTL SNPs as instrument variables have also generally not observed significant associations between LTL wGRS and breast cancer^{18, 19}. One possible explanation for the discordant findings with genetically determined LTL (wGRS) and breast cancer association we observed in our study

could be due to the still relatively modest heritability explained by the known LTL-associated variants. Approximately 4.7% of LTL variance was explained by the LTL wGRS, however, we estimated SNP-based heritability in our study to be approximately 20.6% (Supplementary table 6). With additional larger scale studies and LTL associated variants identified, more comprehensive LTL wGRS may serve as better instruments to evaluate the role of telomere length with breast cancer.

“At the same time, we note on the discordant lack of association between our LTL wGRS and breast cancer in our study as compared to recent reports indicating that longer LTL was associated with increased risks of breast cancer²⁰. It should be noted that the known LTL-associated variants used to generate the wGRS still only explain a modest proportion of phenotypic variance (approximately 4.7%, Supplementary table 4, as compared to the overall SNP-based heritability of approximately 20.6% in our study, Supplementary table 6) and improved genetic instruments may be required to firmly evaluate the role of telomere length in breast cancer.” (Discussion, Page 20, line 397-403).

8. Interpretation of wGRS association with lung cancer. Results in Supplementary Table 15 suggest that the association is also driven by variation at the *TERC* locus, along with that at the *TERT* locus.

Response: We thank the reviewer for pointing this out and we have modified this accordingly.

“...lung cancer associations were driven mainly through rs7705526 at the *TERT* gene locus and rs2293607 at the *TERC* gene locus (Supplementary table 20).” (Discussion, Page 19, line 392-393)

9. Overstated conclusions. In this study, only one low frequency variant was associated with colon cancer risk. Therefore, it seems to be a huge stretch to conclude the following in the abstract: “Ethnic-specific and potentially pathogenic coding and non-coding regulatory low frequency variants contribute substantially to LTL homeostasis in East Asians.”

Response: As recommended by the reviewer, we have modified this conclusion to state:

“Ethnicity-specific low frequency variants may affect LTL homeostasis in East Asians.” (Abstract, Page 3, line 14-15)

“In conclusion, our evaluation of East Asian samples identified low frequency variants that contribute to LTL homeostasis.” (Conclusion, Page 21, line 420-421)

Minor comments:

1. For consistency and clarity, the wording “LTL-associated variants” over “LTL risk variants” is preferable.

Response: We thank the reviewer for highlighting this and have replaced “LTL-associated variants” with “LTL risk variants” in the manuscript.

“Subsequently, LTL-associated variants were evaluated for associations with mortality and incident cancers” (Page 5, line 51-52)

“Genome-wide LTL-associated variants in the Singapore Chinese.” (Page 13, line 239)

“Replication of known LTL-associated low frequency variants.” (Page 16, line 305)

“It should be noted that the known LTL-associated variants used to generate the wGRS.....” (Page 20, line 399-400)

2. The authors refer to “LTL levels” in describing certain results. However, the methods do not define any levels. Please clarify.

Response: We thank the reviewer for highlighting this. We had used “LTL levels” to refer to telomere length as well. We have excluded “levels” and changed “LTL levels” to “LTL” in the revised manuscript to avoid confusion.

3. Suggest replacing the label “controls” to “non-cases” in Table 2, since Cox (and not logistic) regression analyses were performed.

Response: We thank the reviewer for this suggestion and have changed “controls” to “non-cases” in all the tables and supplementary tables.

4. Suggest presenting the results of associations by histology for lung cancer in the main tables (i.e., Table 3), instead of a supplementary table.

Response: We thank the reviewer for the suggestion and have moved the results of associations by histology for lung cancer to Table 4.

References:

1. Wang F, Podell ER, Zaugg AJ, et al. The POT1-TPP1 telomere complex is a telomerase processivity factor. *Nature*. 2007; 445: 506-10.
2. Gu P, Wang Y, Bisht KK, et al. Pot1 OB-fold mutations unleash telomere instability to initiate tumorigenesis. *Oncogene*. 2017; 36: 1939-51.
3. Wu Y, Poulos RC and Reddel RR. Role of POT1 in Human Cancer. *Cancers (Basel)*. 2020; 12.
4. Yachdav G, Kloppmann E, Kajan L, et al. PredictProtein--an open resource for online prediction of protein structural and functional features. *Nucleic Acids Res*. 2014; 42: W337-43.
5. Calvete O, Garcia-Pavia P, Domínguez F, et al. The wide spectrum of POT1 gene variants correlates with multiple cancer types. *Eur J Hum Genet*. 2017; 25: 1278-81.
6. Rice C, Shastrula PK, Kossenkov AV, et al. Structural and functional analysis of the human POT1-TPP1 telomeric complex. *Nat Commun*. 2017; 8: 14928.
7. Shen E, Xiu J, Lopez GY, et al. POT1 mutation spectrum in tumour types commonly diagnosed among POT1-associated hereditary cancer syndrome families. *J Med Genet*. 2020; 57: 664-70.
8. Schlessinger A and Rost B. Protein flexibility and rigidity predicted from sequence. *Proteins*. 2005; 61: 115-26.
9. Dorajoo R, Sun Y, Han Y, et al. A genome-wide association study of n-3 and n-6 plasma fatty acids in a Singaporean Chinese population. *Genes Nutr*. 2015; 10: 53.
10. Han Y, Dorajoo R, Ke T, et al. Interaction effects between Paraoxonase 1 variants and cigarette smoking on risk of coronary heart disease in a Singaporean Chinese population. *Atherosclerosis*. 2015; 240: 40-5.
11. Chang X, Salim A, Dorajoo R, et al. Utility of genetic and non-genetic risk factors in predicting coronary heart disease in Singaporean Chinese. *Eur J Prev Cardiol*. 2017; 24: 153-60.
12. Chang X, Dorajoo R, Sun Y, et al. Gene-diet interaction effects on BMI levels in the Singapore Chinese population. *Nutr J*. 2018; 17: 31.
13. Kapiteijn E, Liefers GJ, Los LC, et al. Mechanisms of oncogenesis in colon versus rectal cancer. *J Pathol*. 2001; 195: 171-8.
14. Frattini M, Balestra D, Suardi S, et al. Different genetic features associated with colon and rectal carcinogenesis. *Clin Cancer Res*. 2004; 10: 4015-21.
15. Zaitlen N, Kraft P, Patterson N, et al. Using extended genealogy to estimate components of heritability for 23 quantitative and dichotomous traits. *PLoS Genet*. 2013; 9: e1003520.
16. Yang J, Lee SH, Goddard ME and Visscher PM. GCTA: a tool for genome-wide complex trait analysis. *Am J Hum Genet*. 2011; 88: 76-82.
17. Ishigaki K, Akiyama M, Kanai M, et al. Large-scale genome-wide association study in a Japanese population identifies novel susceptibility loci across different diseases. *Nat Genet*. 2020; 52: 669-79.
18. Haycock PC, Burgess S, Nounu A, et al. Association Between Telomere Length and Risk of Cancer and Non-Neoplastic Diseases: A Mendelian Randomization Study. *JAMA Oncol*. 2017; 3: 636-51.
19. Zhang C, Doherty JA, Burgess S, et al. Genetic determinants of telomere length and risk of common cancers: a Mendelian randomization study. *Hum Mol Genet*. 2015; 24: 5356-66.
20. Samavat H, Xun X, Jin A, Wang R, Koh WP and Yuan JM. Association between prediagnostic leukocyte telomere length and breast cancer risk: the Singapore Chinese Health Study. *Breast Cancer Res*. 2019; 21: 50.

REVIEWERS' COMMENTS:

Reviewer #1 (Remarks to the Author):

The authors have presented a thoroughly revised manuscript assessing the role of rare variants in altering leukocyte telomere length and contributing to cancer risk/mortality, and have now included additional data on CAD. The updated heritability analyses that I requested are very helpful and appropriately performed. I thank the authors for making the effort to do these analyses, which are not trivial. The revised text improves the clarity and presentation of results and the authors are to be commended. The POT1 association remains unusual, but the authors have done due diligence in performing substantial *in silico* modeling of this mutation. Short of using CRISPR or TALENS to put the mutation in a LCL model (beyond the scope of this paper), there's really no way to further evaluate this mutation. Such is the nature of science and I suspect future molecular research will address this puzzle. I congratulate the authors on their fine study and their responsiveness to my comments and recommend this paper for publication in *Comms Bio*.

Sincerely,
Kyle Walsh, PhD (Reviewer 1)
Associate Professor of Neurosurgery
Director of Neuro-epidemiology
Duke University School of Medicine

Reviewer #3 (Remarks to the Author):

The authors present a revised manuscript describing their efforts to identify rare genetic variants associated with leukocyte telomere length (LTL) and assess whether LTL-associated variants are associated with risk for incident cancers, coronary artery disease (CAD), and all-cause and disease-specific mortality. Appropriate analyses were performed to address reviewer comments, with the main results unchanged. I thank the authors for carefully considering and responding to all reviewer comments. My only concern is that no clear rationale or context is provided for now examining associations of LTL-associated variants with CAD and mortality. The introduction remains framed around telomere homeostasis and cancer predisposition, and the manuscript title also indicates this focus. As a result, the manuscript is not as cohesive and focused as it could be.

We thank the reviewers to re-evaluating our manuscript and for the generally positive comments. Additional suggestions provided by the reviewers and editors have been considered and incorporated into the revised version of the manuscript. Details of changes made to the manuscript are indicated below.

REVIEWERS' COMMENTS:

Reviewer #1 (Remarks to the Author):

The authors have presented a thoroughly revised manuscript assessing the role of rare variants in altering leukocyte telomere length and contributing to cancer risk/mortality, and have now included additional data on CAD. The updated heritability analyses that I requested are very helpful and appropriately performed. I thank the authors for making the effort to do these analyses, which are not trivial. The revised text improves the clarity and presentation of results and the authors are to be commended. The POT1 association remains unusual, but the authors have done due diligence in performing substantial in silico modeling of this mutation. Short of using CRISPR or TALENS to put the mutation in a LCL model (beyond the scope of this paper), there's really no way to further evaluate this mutation. Such is the nature of science and I suspect future molecular research will address this puzzle. I congratulate the authors on their fine study and their responsiveness to my comments and recommend this paper for publication in Comms Bio.

Sincerely,

Kyle Walsh, PhD (Reviewer 1)
Associate Professor of Neurosurgery
Director of Neuro-epidemiology
Duke University School of Medicine

Response: We thank Dr. Kyle Walsh (Reviewer 1) for the review of our manuscript and for the positive comments.

Reviewer #3 (Remarks to the Author):

The authors present a revised manuscript describing their efforts to identify rare genetic variants associated with leukocyte telomere length (LTL) and assess whether LTL-associated variants are associated with risk for incident cancers, coronary artery disease (CAD), and all-cause and disease-specific mortality. Appropriate analyses were performed to address reviewer comments, with the main results unchanged. I thank the authors for carefully considering and responding to all reviewer

comments. My only concern is that no clear rationale or context is provided for now examining associations of LTL-associated variants with CAD and mortality. The introduction remains framed around telomere homeostasis and cancer predisposition, and the manuscript title also indicates this focus. As a result, the manuscript is not as cohesive and focused as it could be.

Response: We thank Reviewer 2 for the reassessing our manuscript and for the suggestions. To improve the rationale to evaluate (mortalities and CAD) in our study, we have now provided additional context on telomere length associations with mortalities and other chronic diseases (for eg CAD) in the revised version of the manuscript. The title has also been modified to “Low frequency variants associated with leukocyte telomere length in the Singapore Chinese population” to keep within the ~15 word limit, as recommended. We have also indicated that as the *TERF1* rs79617270 showed a significant association with cancer mortality in our study, we subsequently also evaluated the associations with various incident cancers. The modifications made to the manuscript are:

“The role of low frequency variants associated with telomere length homeostasis in chronic diseases and mortalities is relatively understudied in the East-Asian population.” (Abstract, Page 3, Line 1-2)

“Rs79617270 is associated with cancer mortality [$HR_{95\%CI} = 1.544 (1.173, 2.032)$, $P_{Adj}=0.018$] and 4.76% of the association between the rs79617270 and colon cancer is mediated through LTL.” (Abstract, Page 3, Line 9-10)

“Telomere length have been associated with risks of cancers, other chronic diseases and mortalities¹⁻⁴. Shortening of telomere length leads to an increased risk of acceleration of age-related diseases such as cardiovascular disease^{1,2,5}.” (Introduction, Page 4, Line 15-17)

“However, little is known if such rare variants involved in telomere homeostasis exist in the East Asian population and if these in turn affect predispositions to chronic diseases and mortalities.” (Introduction, Page 4-5, Line 36-38)

“We first investigated whether the low frequency variants identified in the current study were associated with mortalities. Only for cancer mortality we observed that the minor G allele of rs79617270 at *TERF1* region, which was associated with increased telomere length, was associated with increased cancer mortality [$HR_{95\%CI} = 1.544 (1.173, 2.032)$, Cox regression $P_{Adj}=0.018$] (Table 2)..... We further evaluated if the three low frequency variants were associated with incident cancers.” (Results, Page 9, Line 133-141)

References

1. Haycock PC, Burgess S, Nounu A, et al. Association Between Telomere Length and Risk of Cancer and Non-Neoplastic Diseases: A Mendelian Randomization Study. *JAMA Oncol.* 2017; 3: 636-51.
2. Zhan Y, Karlsson IK, Karlsson R, et al. Exploring the Causal Pathway From Telomere Length to Coronary Heart Disease: A Network Mendelian Randomization Study. *Circ Res.* 2017; 121: 214-9.
3. Willeit P, Willeit J, Mayr A, et al. Telomere length and risk of incident cancer and cancer mortality. *JAMA.* 2010; 304: 69-75.
4. Dorajoo R, Chang X, Gurung RL, et al. Loci for human leukocyte telomere length in the Singaporean Chinese population and trans-ethnic genetic studies. *Nature communications.* 2019; 10: 2491.
5. Blackburn EH, Greider CW and Szostak JW. Telomeres and telomerase: the path from maize, Tetrahymena and yeast to human cancer and aging. *Nat Med.* 2006; 12: 1133-8.